# ENHANCING GRAPH SELF-SUPERVISED LEARNING WITH GRAPH INTERPLAY

## ABSTRACT

Graph self-supervised learning (GSSL) has emerged as a compelling framework for extracting informative representations from graph-structured data without extensive reliance on labeled inputs. In this study, we introduce Graph Interplay (GIP), an innovative and versatile approach that significantly enhances the performance equipped with various existing GSSL methods. To this end, GIP advocates direct graph-level communications by introducing random inter-graph edges within standard batches. Against GIP's simplicity, we further theoretically show that GIP essentially performs a principled manifold separation via combining inter-graph message passing and GSSL, bringing about more structured embedding manifolds and thus benefits a series of downstream tasks. Our empirical study demonstrates that GIP surpasses the performance of prevailing GSSL methods across multiple benchmarks by significant margins, highlighting its potential as a breakthrough approach. Besides, GIP can be readily integrated into a series of GSSL methods and consistently offers additional performance gain. This advancement not only amplifies the capability of GSSL but also potentially sets the stage for a novel graph learning paradigm in a broader sense. GIP is open-sourced at `https://anonymous.4open.science/r/GIP`.

## 1 INTRODUCTION

Graph-structured data has become increasingly prevalent across a variety of domains, presenting both unique challenges and opportunities for machine learning innovations. The complexity and irregular nature of graph data, characterized by its intricate relationships and diverse structures, necessitate specialized learning approaches. Graph Self-Supervised Learning (GSSL) has emerged as a pivotal strategy in this context (Jin et al., 2020; Liu et al., 2022; Xie et al., 2022; Wu et al., 2021), enabling the utilization of unlabeled graph data effectively in sectors as wide-ranging as molecular property prediction (Rong et al., 2020; Zhang et al., 2021b; Liu et al., 2021), and recommendation systems (Wu et al., 2021; Yu et al., 2022). The strength of GSSL lies in its capacity to autonomously discover complex patterns and structures within data, a process that is inherently valuable in understanding and exploiting the rich connectedness inherent within graph data.

Despite the promise and advancements in GSSL, much of its development has been influenced by methodologies and ideas borrowed from the domains of computer vision and natural language processing (Chen et al., 2020; He et al., 2020; Devlin et al., 2018). Techniques such as contrastive learning, commonly used loss functions like InfoNCE (Gutmann & Hyvärinen, 2010), Jensen-Shannon estimator (JSE) (Nowozin et al., 2016), and Barlow Twins loss (Zbontar et al., 2021) , data augmentation strategies (Takahashi et al., 2019; Zhang, 2017), as well as specific architecture designs (Grill et al., 2020; He et al., 2022; Liu et al., 2023), have been adapted to fit the graph learning paradigm (You et al., 2020; Hassani & Khasahmadi, 2020; Bielak et al., 2022; Rong et al., 2019; Wu et al., 2022; Thakoor et al., 2021; Hou et al., 2022; Gong et al., 2024; Zhao et al., 2024). While these adaptions have spurred progress, they often overlook the peculiar and critical characteristics of graph data, such as its non-uniformity, the varying connectivity of different nodes, and the complexity of their relational linkages.

The limitations of current GSSL methodologies highlight an urgent need for approaches that are specifically tailored to respect and leverage the unique attributes of graph structures. Conventional methods often fail to tap into the full depth of information available, restricted by their partial

adaptation of techniques from other fields. This realization has directed our research toward exploring novel avenues in graph learning that honor the intrinsic properties of graphs more holistically.

Motivated by these challenges, we have developed Graph Interplay (GIP), a novel conceptual and computational framework designed to enhance the capability of GSSL. GIP introduces an innovative mechanism that integrates random inter-graph edges within batches, facilitating a richer and more dynamic interplay of information across different graphs. This approach is specifically advantageous in the context of GNNs (Graph Neural Networks), which leverage message-passing mechanisms to process graph-structured data. By interconnecting graphs within learning batches, GIP effectively broadens the contextual landscape within which the learning model operates, thus allowing for a more comprehensive understanding of manifold structures across diverse graph examples.

Theoretically, we show that GIP equipped with GNNs provides a platform for better manifold discovery and separation in the realm of graph data, a critical aspect in enhancing the quality and applicability of learned representations. This theoretical basis underpins the practical benefits of GIP, demonstrating how it offers more discriminating and informative graph representations that are likely to improve performance on downstream tasks. Empirically, we applied GIP to a range of GSSL frameworks and noted significant improvements across multiple benchmarks, as shown in Figure 1. For instance, in challenging graph classification datasets like IMDB-MULTI, the incorporation of GIP elevated the classification accuracy from sub-60% levels to over 90%, showcasing its efficacy and potential as an innovative paradigm in GSSL.

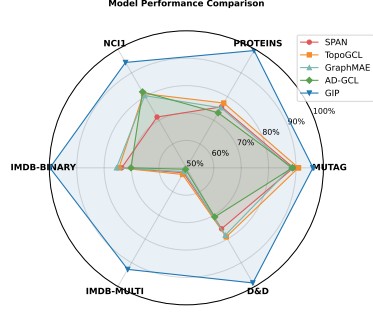

Figure 1: Performance comparison of GSSL methods.

The contributions of this paper articulate the core innovations and advancements offered by GIP: **(I)** We introduce Graph Interplay (GIP), a ground-breaking enhancement to graph self-supervised learning that encourages effective inter-graph connectivity for enriched learning experiences. **(II)** We make a step to provide a theoretical foundation for understanding GIP, elucidating its potential for improved manifold separation within graph domains. **(III)** We validate the effectiveness of GIP through comprehensive empirical studies across a diverse range of graph-level benchmarks, where GIP has shown remarkable improvements and versatility, significantly elevating the performance metrics of existing GSSL setups.

## 2 RELATED WORK

**Graph Self-Supervised Learning (GSSL).** GSSL methods can be categorized into Graph Contrastive Learning (GCL) and Graph Predictive Learning (Xie et al., 2022). GCL employs augmentations to create multiple views of the input graph, learning to maximize mutual information between these views for robust and invariant representations. Typically, GCL approaches typically focus on maximizing a lower bound of mutual information using estimators like InfoNCE (Gutmann & Hyvärinen, 2010), and JSE (Nowozin et al., 2016). Examples of frameworks utilizing the InfoNCE objective include GRACE (Zhu et al., 2020), GCC (Qiu et al., 2020), and GCA (Zhu et al., 2021b), while MVGRL (Hassani & Khasahmadi, 2020) and InfoGraph (Sun et al., 2019) employ JSE. Predictive learning methods train graph encoders using self-generated labels and prediction heads. These include graph autoencoder-based models like GAE (Kipf & Welling, 2016b), MGAE (Wang et al., 2017), GALA (Park et al., 2019),VGAE (Kipf & Welling, 2016b), and ARGA/ARVGA (Pan et al., 2018), which capture representations through reconstruction. Additionally, models such as $S^2$GRL (Peng et al., 2020) and GROVER (Rong et al., 2020) predict specific statistical properties associated with the graph, further enhancing their ability to learn meaningful representations. Other methods like M3S (Sun et al., 2020) and ICF-GCN (Hu et al., 2021) utilize self-training and node clustering for self-supervised signals. Furthermore, approaches such as BGRL (Thakoor et al., 2021) and CCA-SSG (Zhang et al., 2021a) achieve robust learning through invariance regularization, eliminating the need for negative sample pairs.

**Manifold Perspective on Self-Supervised Learning.** Based on the manifold hypothesis, which posits that high-dimensional data often lies on low-dimensional manifolds, SSL can be viewed as learning the structure of these underlying manifolds (Bengio et al., 2013). Recent approaches in analyzing SSL

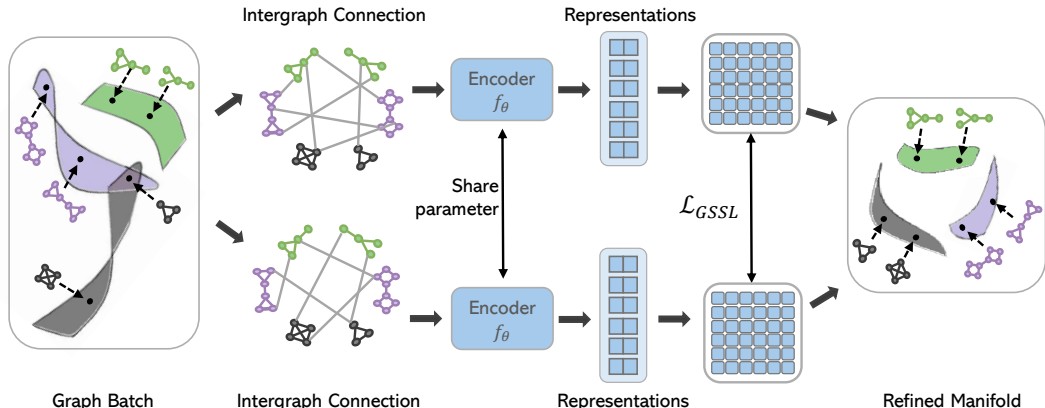

Figure 2: Overview of the GIP framework. Individual graphs are stochastically interconnected to form enriched views. These views allow each instance to perceive a rich topological context through the shared GNN encoder, enabling GSSL to leverage enhanced structural information for learning graph representations.

from a manifold perspective often start by viewing relationship graphs as discrete approximations of the data manifolds. These graphs are typically constructed by defining edges based on sample similarities (Balestriero & LeCun, 2022; Munkhoeva & Oseledets, 2024) or augmentations (HaoChen et al., 2021). Spectral techniques are then employed to analyze these graph structures. Balestriero & LeCun (2022) established equivalences between SSL methods and spectral embedding techniques like ISOMAP (Balasubramanian & Schwartz, 2002). Tan et al. (2024) proved the equivalence of SimCLR (Chen et al., 2020) and spectral clustering on predefined similarity graphs and designed empirically more powerful comparison learning objectives based on the maximum entropy principle. These theoretical advancements not only deepen our understanding of SSL but also guide the development of more effective algorithms grounded in manifold learning principles.

## 3 METHOD

In this section, we introduce Graph Interplay (GIP), which is designed to enhance GSSL through direct graph-level communications. We begin by outlining the motivation behind GIP, followed by a detailed description of its core mechanism, as well as its integration with existing GSSL frameworks. Finally, we analyze how GIP achieves a better manifold separation and provide theoretical insights into why GIP leads to more effective graph representations.

### 3.1 MOTIVATION

GSSL has emerged as a powerful paradigm for learning representations from graph-structured data without relying on explicit labels. However, current GSSL methods face several limitations: **(I)** Limited Inter-graph Information Exchange: Existing methods typically process graphs independently or rely on indirect interactions through parameter sharing, missing opportunities to leverage broader contextual information across the entire graph set. **(II)** Inefficient Use of Batch Information: Although graphs are often processed in batches, the structural information within a batch is not fully utilized, leaving the potential for graphs to inform and enhance each other's representations largely untapped. **(III)** Constrained View Generation: Most existing augmentation techniques focus on intra-graph operations, which may not capture the full spectrum of graph variations present in the data, potentially limiting the model's ability to learn robust and generalizable representations. These limitations collectively restrict the ability of current GSSL methods to fully capture and leverage the rich, complex dependencies that often exist within graph-structured data, potentially hindering their performance on downstream tasks.

### 3.2 OVERVIEW

The GIP process integrates seamlessly with existing GSSL schemes and can be summarized as follows: **(I)** Batch Sampling: A batch of graphs is sampled from a collection of pre-processed graphs.

**(II)** Inter-graph Edge Addition: GIP randomly adds edges between graphs in the batch, creating two distinct views. These added edges establish message-passing channels between graphs, allowing for information flow across the batch. **(III)** Representation readout: Each graph in these two views now has access to a broader range of structural information. The GNN encoder and pooling function process this expanded structure, fusing information from both the original graph and the introduced inter-graph interplay. **(IV)** GSSL-driven Representation Learning: Graph representations from the two views are used to compute pairwise similarity matrices. These matrices serve as input to various GSSL objectives, including contrastive and invariance-keeping reduction methods. This flexibility allows GIP to integrate with a wide range of GSSL methods, guiding the learning process to capture meaningful patterns and relationships within the enriched graph structures. The framework of GIP is outlined in Figure 2.

## 3.3 GRAPH INTERPLAY (GIP)

To address the limitations of existing GSSL methods, we propose Graph Interplay (GIP), a novel approach that fundamentally reimagines how graphs interact during the self-supervised learning process. GIP transcends the conventional view of graphs as isolated entities, instead conceptualizing them as interconnected components of a larger, dynamic system. The core innovation of GIP lies in its ability to create enhanced views of the graph dataset through the strategic introduction of stochastic inter-graph edges. This process transforms a batch of disparate graphs into a unified, information-rich structure. For frameworks requiring two views, GIP can generate these using two independent probability parameters. Given a batch of graphs $\mathcal{G} = \{\mathcal{G}_1, \mathcal{G}_2, ..., \mathcal{G}_N\}$, where each graph $\mathcal{G}_i = (\mathcal{V}_i, \mathcal{E}_i)$, GIP introduces stochastic inter-graph edges to create an extended edge set:

$$\mathcal{E}_{\text{ext}} = \bigcup_{i=1}^{N} \mathcal{E}_i \cup \mathcal{E}_{\text{inter}}, \quad P((u,v) \in \mathcal{E}_{\text{inter}}) = p \quad \text{if } u \in \mathcal{V}_i, v \in \mathcal{V}_j, i \neq j \tag{1}$$

Here, $\mathcal{E}_{\text{ext}}$ represents the extended edge set, $\mathcal{E}_{\text{inter}}$ denotes the set of inter-graph edges, $p$ is the probability of adding an inter-graph edge. For GSSL frameworks that require two views, we can generate these by assigning two independent probabilities $p_1$ and $p_2$, each used to create a separate instance of $\mathcal{E}_{\text{ext}}$.

The GIP-enhanced message passing process operates on this extended graph structure. For each node $v$, its representation is updated as:

$$\mathbf{h}_v^{(l+1)} = \text{UPDATE}^{(l)} \left( \mathbf{h}_v^{(l)}, \text{AGGR}^{(l)} \left( \left\{ \text{MSG}^{(l)}(\mathbf{h}_v^{(l)}, \mathbf{h}_u^{(l)}) : (u,v) \in \mathcal{E}_{\text{ext}} \right\} \right) \right) \tag{2}$$

In this equation, $\mathbf{h}_v^{(l)}$ denotes the representation of node $v$ at layer $l$. The function $\text{MSG}^{(l)}$ computes the message from a neighbor node $u$ to node $v$, $\text{AGGR}^{(l)}$ aggregates messages from all neighbors, and $\text{UPDATE}^{(l)}$ produces the new node representation. This formulation allows each node to assimilate information from a diverse, dynamically generated context spanning multiple graphs, providing a unique perspective on the inter-graph relationships.

After $L$ layers of message passing, we obtain graph-level representations through a pooling operation:

$$\mathbf{h}_{G_i} = \text{POOL}(\{\mathbf{h}_v | v \in \mathcal{V}_i\}) \tag{3}$$

where $\mathbf{h}_{\mathcal{G}_i} \in \mathbb{R}^d$ is the graph-level representation for $\mathcal{G}_i$, and POOL is a pooling function that aggregates node representations into a single graph representation.

## 3.4 INTEGRATION WITH GSSL FRAMEWORKS

The stochastic nature of GIP's inter-graph connections serves a dual purpose. First, it acts as an implicit regularizer, preventing overfitting to specific graph structures. Second, it generates a rich set of graph views, addressing the limited view generation problem of traditional augmentation techniques. GIP is designed to be integrated into various self-supervised learning objectives, including both contrastive and redundancy-reduction methods. The specific formulation of these objectives can vary depending on the chosen framework. For a detailed discussion of how GIP can be incorporated into different self-supervised learning objectives, we refer the reader to Appendix C.

By applying GIP during the pretraining stage, we fundamentally alter the learning dynamics of GSSL. Graphs no longer learn in isolation, but instead engage in a collaborative learning process, sharing insights and co-evolving their representations. This collective learning approach enables the model to capture higher-order structures and relationships that are invisible when processing graphs independently.

## 3.5 Relation to Manifold Separation

In this section, we formally analyze how GIP enhances manifold separation in the representation space, leading to improved graph representation learning. To bridge the gap between the practical implementation of GIP and our theoretical analysis, we introduce simplifying assumptions and definitions that capture the essence of GIP while making the problem mathematically tractable. We consider a set of graphs $\mathcal{S} = \{\mathcal{G}_1, \mathcal{G}_2, \ldots, \mathcal{G}_N\}$ lying on $K$ underlying manifolds $\mathcal{F} = \{\mathcal{M}_1, \mathcal{M}_2, \ldots, \mathcal{M}_K\}$ in a high-dimensional space. Each manifold $\mathcal{M}_k$ is associated with a probability distribution $P_k$ from which graphs are sampled. This abstraction allows us to model the inherent structure of the graph dataset and analyze how GIP affects the relationships between graphs from the same or different manifolds. To capture the essence of GIP's inter-graph communication mechanism, we propose the following lemma:

**Lemma 1** (GIP Transformation). *Consider a GNN with $n$ layers ($n \geq 1$) used in Graph Interplay (GIP), under the following conditions:*

- *Each layer of the GNN consists of a linear transformation followed by a ReLU activation function.*

- *The pooling operation used to obtain graph-level representations is additive.*

*Then the GIP transformation can be equivalently represented as:*

$$f_g(\mathcal{G}_i) = f(\mathcal{G}_i) + \sum_{j \neq i} \alpha_{ij} f(\mathcal{G}_j) \tag{4}$$

*where $f : \mathcal{G} \to \mathbb{R}^d$ is a GNN encoder, and $\alpha_{ij}$ are learnable parameters representing the strength of interaction between graphs $\mathcal{G}_i$ and $\mathcal{G}_j$.*

This formulation abstracts GIP into a more compact form, facilitating our theoretical analysis of its impact on manifold separation. The proof of this lemma can be found in the Appendix G.1. To quantify the effectiveness of GIP in separating manifolds, we introduce the concept of manifold-relevant information $Z_k$ as a random variable for each manifold:

$$Z_k = f_s(\mathcal{G}), \quad \mathcal{G} \sim P_k \tag{5}$$

where $P_k$ is the probability distribution over graphs in manifold $\mathcal{M}_k$, and $f_s$ denotes the GNN encoder that has been well-trained through standard SSL. This formulation allows us to measure GIP's enhancement in manifold alignment and separation over standard SSL. With these definitions in place, we can now state our main theoretical result:

**Theorem 1** (GIP's Improvement on Manifold Separation). *Given the above definitions and assumptions, under the self-supervised learning objective and sufficient training, GIP can achieve better expected manifold separation than SSL:*

$$\frac{\mathbb{E}_{\mathcal{G}_i \sim P_k}[I(f_g^{(v)}(\mathcal{G}_i); Z_k)]}{\max_{l \neq k} \mathbb{E}_{\mathcal{G}_i \sim P_k}[I(f_g^{(v)}(\mathcal{G}_i); Z_l)]} > \frac{\mathbb{E}_{\mathcal{G}_i \sim P_k}[I(f_s(\mathcal{G}_i); Z_k)]}{\max_{l \neq k} \mathbb{E}_{\mathcal{G}_i \sim P_k}[I(f_s(\mathcal{G}_i); Z_l)]}, \quad v \in \{1, 2\} \tag{6}$$

*where $I(\cdot; \cdot)$ denotes mutual information and $f_g^{(v)}$ represents the GIP embedding function for view $v$.*

This theorem formalizes the intuition that GIP enhances the separation between manifolds in the representation space in both views. By analyzing how the self-supervised learning objective interacts with the inter-graph information exchange process, we show that GIP systematically increases the ratio of intra-manifold information to inter-manifold information. Specifically, GIP enhances intra-manifold similarities while keeping inter-manifold similarities constant, leading to more discriminative representations. Our theoretical analysis provides a conservative estimate of GIP's potential.

Table 1: Graph classification. MVGRL+PPR is the original setting of MVGRL. The best results in each cell are highlighted by grey . The best results overall are highlighted with **bold and underline.** Metric is accuracy (%).

| Model | MUTAG | PROTEINS | NCI1 | IMDB-BINARY | IMDB-MULTI | DD |
|---|---|---|---|---|---|---|
| GraphCL | 86.80 ± 1.34 | 74.39 ± 0.45 | 77.87 ± 0.41 | 71.14 ± 0.44 | 48.58 ± 0.67 | 78.62 ± 0.40 |
| AD-GCL | 88.74 ± 1.85 | 73.28 ± 0.46 | 82.00 ± 0.29 | 70.21 ± 0.68 | 50.60 ± 0.70 | 75.79 ± 0.87 |
| RGCL | 87.66 ± 1.01 | 75.03 ± 0.43 | 78.14 ± 1.08 | 71.85 ± 0.84 | 49.31 ± 0.42 | 78.86 ± 0.48 |
| SPAN | 89.12 ± 0.76 | 75.78 ± 0.41 | 71.43 ± 0.49 | 73.65 ± 0.69 | 52.16 ± 0.72 | 75.78 ± 0.52 |
| GraphMAE | 88.19 ± 1.26 | 75.30 ± 0.39 | 80.40 ± 0.30 | 75.52 ± 0.66 | 51.63 ± 0.52 | 78.47 ± 0.23 |
| TopoGCL | 90.09 ± 0.93 | 77.30 ± 0.89 | 81.30 ± 0.27 | 74.67 ± 0.32 | 52.81 ± 0.31 | 79.15 ± 0.35 |
| MVGRL + PPR | 90.00 ± 5.40 | 78.92 ± 1.83 | 78.78 ± 1.52 | 71.40 ± 4.17 | 52.13 ± 1.42 | 88.38 ± 0.31 |
| MVGRL+ DROPEDGE | 93.33 ± 5.44 | 82.34 ± 2.59 | 75.52 ± 1.13 | 70.00 ± 2.61 | 50.40 ± 2.82 | 85.47 ± 0.94 |
| MVGRL+ ADDEDGE | 94.44 ± 0.00 | 87.57 ± 1.55 | 82.09 ± 0.88 | 75.00 ± 4.98 | 53.47 ± 3.14 | 94.02 ± 1.52 |
| **MVGRL + GIP** | **96.27 ± 2.72** | 98.20 ± 0.74 | 92.02 ± 1.92 | 92.67 ± 2.87 | 69.73 ± 5.05 | 98.58 ± 0.81 |
| G-BT + DROPEDGE | 92.59 ± 2.61 | 77.97 ± 0.42 | 78.18 ± 0.91 | 73.33 ± 1.24 | 49.11 ± 1.25 | 78.29 ± 1.99 |
| G-BT + ADDEDGE | 92.59 ± 2.61 | 80.64 ± 1.68 | 75.91 ± 0.59 | 73.33 ± 1.24 | 48.88 ± 1.13 | 81.03 ± 1.98 |
| **G-BT + GIP** | 92.59 ± 5.24 | 98.20 ± 1.27 | **94.64 ± 0.60** | 81.67 ± 3.30 | 64.44 ± 4.01 | 96.92 ± 1.12 |
| BGRL + DROPEDGE | 91.11 ± 2.72 | 78.02 ± 0.72 | 74.70 ± 0.69 | 74.20 ± 1.72 | 47.74 ± 3.23 | 80.68 ± 2.45 |
| BGRL + ADDEDGE | 87.78 ± 5.44 | 84.68 ± 3.86 | 80.34 ± 2.15 | 76.00 ± 2.28 | 47.47 ± 1.86 | 90.26 ± 1.59 |
| **BGRL + GIP** | 92.59 ± 1.52 | 97.84 ± 1.35 | 83.45 ± 0.75 | **99.80 ± 0.40** | 92.00 ± 1.52 | 97.44 ± 1.69 |
| GRACE + DROPEDGE | 88.89 ± 4.97 | 82.34 ± 0.92 | 74.45 ± 1.12 | 69.20 ± 2.56 | 46.00 ± 1.74 | 79.49 ± 2.42 |
| GRACE + ADDEDGE | 92.22 ± 4.44 | 86.13 ± 2.32 | 83.02 ± 1.06 | 68.60 ± 2.42 | 46.80 ± 0.88 | 84.79 ± 1.90 |
| **GRACE + GIP** | 91.11 ± 5.67 | **99.40 ± 0.85** | 94.00 ± 0.61 | 99.33 ± 0.47 | **92.89 ± 3.19** | **98.58 ± 0.81** |

Table 2: Results on the graph-level tasks. ↓ means lower the better, and ↑ means higher the better.

| Task Dataset | Regression (Metric: RMSE ↓) | | | Classification (Metric: ROC-AUC% ↑) | | |
|---|---|---|---|---|---|---|
| | molesol | mollipo | molfreesolv | molbace | molbbbp | molclintox |
| InfoGraph | 1.344±0.178 | 1.005±0.023 | 10.005±4.819 | 74.74±3.64 | 66.33±2.79 | 64.50±5.32 |
| GraphCL | 1.272±0.089 | 0.910±0.016 | 7.679±2.748 | 74.32±2.70 | 68.22±1.89 | 74.92±4.42 |
| JOAO | 1.285±0.121 | 0.865±0.032 | 5.131±0.722 | 74.43±1.94 | 67.62±1.29 | 78.21±4.12 |
| AD-GCL | 1.217±0.087 | 0.842±0.028 | 5.150±0.624 | 76.37±2.03 | 68.24±1.47 | 80.77±3.92 |
| SPAN | 1.218±0.052 | 0.802±0.019 | 4.531±0.463 | 76.74±2.02 | 69.59±1.34 | 80.28±2.42 |
| Sp$^2$GCL | 1.235±0.119 | 0.835±0.026 | 4.144±0.573 | 78.76±1.43 | 68.72±1.53 | 80.88±3.86 |
| MVGRL | 1.303 ± 0.135 | 0.958 ± 0.158 | 2.467 ± 0.377 | 77.28 ± 2.13 | 68.31 ± 1.02 | 85.37 ± 3.53 |
| MVGRL + GIP | 1.282 ± 0.059 | 0.948 ± 0.093 | 2.421 ± 0.324 | **91.00 ± 3.25** | 69.12 ± 1.88 | **87.06 ± 2.17** |
| GRACE | 1.358 ± 0.047 | 0.866 ± 0.018 | **2.396 ± 0.228** | 79.40 ± 1.38 | 68.21 ± 1.53 | 86.89 ± 2.39 |
| GRACE + GIP | **1.196 ± 0.061** | 0.805 ± 0.020 | 2.782 ± 0.292 | 87.78 ± 3.93 | **70.92 ± 1.65** | 87.01 ± 2.19 |

In practice, GIP's iterative refinement of representations and enhancement of manifold separation may lead to even more distinctive graph representations. This result offers a formal justification for the empirical success of GIP, demonstrating that its core mechanism of inter-graph communication indeed leads to more effective graph representations. Detailed definitions, assumptions, proof, and further theoretical insights are provided in Appendix G.

# 4 EXPERIMENT

In this section, we conducted a comprehensive evaluation of GIP across 12 datasets, where GIP exhibited notable improvements in the majority of datasets. To further elucidate the factors contributing to GIP's performance, we subsequently performed rigorous analytical experiments, providing deeper insights into its underlying mechanisms.

## 4.1 MAIN RESULTS

**Datasets and Protocols** We test on multiple graph classification and regression datasets ranging from social networks, and chemical molecules to biological networks. We benchmark our model on the TU Datasets (Morris et al., 2020) and OGB graph property prediction datasets (Hu et al., 2020). For both graph classification and regression tasks, we follow the evaluation protocols established in previous works (Lin et al., 2023; Chen et al., 2024a). Specifically, we first train our model in a self-supervised manner to learn graph representations. Then, we freeze the pre-trained encoder and

use it to extract features for downstream tasks. For evaluation, we train a linear classifier or regressor on top of these frozen features and report the performance on the test set. For TU Datasets, we apply 10-fold cross-validation, while for OGB datasets, we use the provided data split. Additional details regarding dataset statistics can be found in the Appendix B.

**Setup and Baselines.** We equip GIP with four Graph SSL frameworks: MVGRL (Hassani & Khasahmadi, 2020), GRACE (Zhu et al., 2020), G-BT (Bielak et al., 2022), and BGRL (Thakoor et al., 2021) following the previous works (Lin et al., 2023). Using DROPEDGE and ADDEDGE as augmentation strategies, details are in Appendix B. For MVGRL, we also compared its original Personalized PageRank (PPR) augmentation (Page, 1998). For the TU Datasets, We compare GIP with six GSSL methods including GraphCL (You et al., 2020), AD-GCL (Suresh et al., 2021), RGCL (Li et al., 2022), SPAN (Lin et al., 2023), GraphMAE (Hou et al., 2022), and TopoGCL (Chen et al., 2024b). For OGB graph property prediction datasets, We compare GIP with six GSSL methods including InfoGraph (Sun et al., 2019), JOAO (You et al., 2021), GraphCL, AD-GCL, SPAN and SP$^2$GCL (Bo et al., 2024). More implementation details can be found in the Appendix B.

**Main results.** Experimental results presented in Table 1 demonstrate that GIP consistently enhances the performance of four different self-supervised learning frameworks: MVGRL, G-BT, GRACE, and BGRL. Across all six datasets, GIP-enhanced models achieve state-of-the-art performance, often surpassing previous methods by a significant margin. Notably, GIP shows substantial improvements on the IMDB-MULTI dataset, where other self-supervised learning methods have struggled to achieve high performance. The consistent improvements across diverse datasets and frameworks align with our theoretical analysis of GIP's ability to enhance intra-manifold mutual information while reducing inter-manifold mutual information. This is evident in the enhanced classification performance, which indicates better separation of graph manifolds in the learned feature space. Interestingly, while the base performance of different frameworks varies, GIP consistently elevates their performance to a similar, high level. This observation supports our theoretical argument that GIP can effectively filter and enhance relevant structural information, regardless of the specific self-supervised learning paradigm employed. The near-perfect classification performance achieved on several datasets further validates our analysis of GIP's capacity to leverage graph interplay for more effective feature learning. These results not only demonstrate the effectiveness of GIP but also its versatility across different self-supervised learning paradigms and dataset characteristics.

We also evaluated the performance of GIP on six chemical molecular property classification and regression tasks in the Open Graph Benchmark. Specifically, we implemented GIP on top of two frameworks, GRACE and MVGRL. Our results demonstrate that GIP consistently and significantly improves performance on five out of six datasets, except for *molfreesolv* dataset. Moreover, GIP remains competitive with state-of-the-art Graph SSL methods, achieving the best results on four datasets, most notably on the *molbace* dataset. Detailed results are reported in Table 2. To investigate the exception, we further analyzed the *molfreesolv* dataset, where GIP did not show improvement. We visualized the performance of GRACE on this dataset with respect to the edge perturbation probability of the two views in Figure 3, using the two-branch GRACE

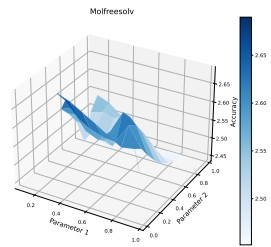

Figure 3: Effect of two-branch DROPEDGE parameters on OGBG-Molfreesolv (RMSE).

framework with DROPEDGE as a data augmentation technique. Interestingly, we found that the *molfreesolv* regression task obtains the best performance when the DROPEDGE probability is close to 1. This implies that *molfreesolv*'s dependence on topology is relatively low, making it difficult for GIP's mechanism to provide significant benefits for this particular dataset.

## 4.2 ABLATION STUDY AND ANALYSIS

**Varying GIP probability.** To systematically investigate the impact of our proposed Graph Interplay (GIP) mechanism on model performance, we conducted a comprehensive experiment varying the edge addition probabilities $(p_1, p_2)$ within the GRACE framework. Figure 4 visualizes the results across multiple datasets from the TUDataset collection as 3D surface plots, where the $x$ and $y$ axes represent $p_1$ and $p_2$ respectively, ranging from 0 to 1, and the $z$-axis represents the achieved accuracy. These visualizations reveal a clear trend: higher proportions of added edges, generally improve model

performance, with peak accuracy typically observed when both $p_1$ and $p_2$ approach 1. This finding suggests that facilitating extensive information exchange between graphs significantly enhances the quality of learned representations. For comparison, we conducted similar visualizations for the DROPEDGE and ADDEDGE methods in Appendix D. Interestingly, these baseline approaches showed highly dataset-dependent behaviors with complex, often non-monotonic relationships between edge manipulation probabilities and accuracy. The clear principles governing GIP's performance offer promising and consistent avenues for further theoretical and empirical exploration, potentially leading to even more effective GSSL techniques.

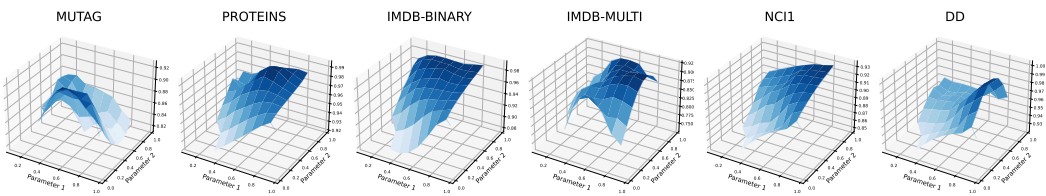

Figure 4: Effect of two-branch GIP parameters on accuracy. A clear trend is that as the proportion of added edges increases, meaning the graphs interplay more frequently, the performance improves.

**GIP with deeper GNNs.** To further investigate the efficacy of GIP, we conducted extensive experiments varying the number of GNN layers in our model. Figure 5 illustrates the performance of GIP compared to baseline graph augmentation methods across different GNN depths on five datasets. The baseline methods include DROPEDGE, ADDEDGE, and Random Walk Sampling (RWS), providing a comprehensive comparison. The results reveal a striking contrast: while GIP consistently benefits from deeper GNN architectures, the baseline methods struggle to leverage increased depth effectively. Specifically, GIP shows a clear upward trend in accuracy as the number of GNN layers increases from 2 to 5 across all datasets, with the most pronounced improvements observed in IMDB-MULTI and IMDB-BINARY. In contrast, baseline methods struggle with increased depth, exhibiting either stagnant performance or degradation, particularly beyond 3 layers. This superior performance of GIP with deeper architectures can be attributed to its ability to effectively utilize expanded receptive fields. As GNN depth increases, the model captures more comprehensive information flows from other graphs, providing richer resources for self-supervised learning and enabling better adjustment of the manifold configuration of learned representations. While conventional methods demonstrate limited effectiveness with deeper architectures, GIP exhibits the potential to unlock the full capacity of deep GNNs in Graph SSL.

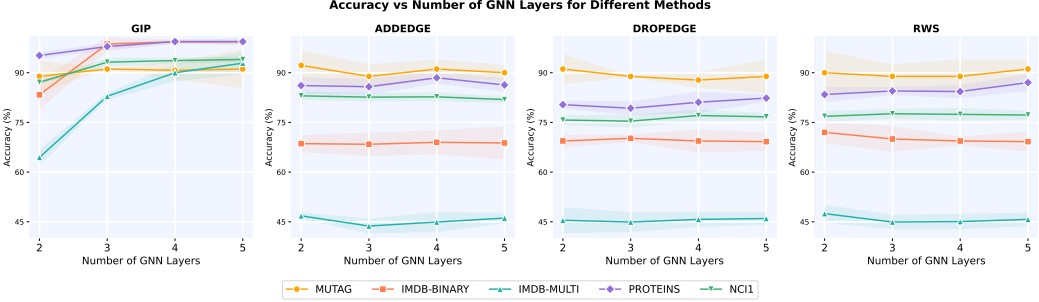

Figure 5: Comparison of accuracy across different numbers of GNN layers for three methods: GIP, ADDEDGE, and DROPEDGE. GIP consistently outperforms the other methods across all datasets, showing a general trend of improved accuracy with increased layer depth.

**Effect of different starting layers of GIP.** To further understand the impact of our Graph Interplay mechanism, we conducted experiments to investigate the effect of applying GIP at different depths within the GNN architecture. In this context, the starting layer refers to the GNN layer from which we begin to apply GIP, with earlier layers using the original graph topology. Figure 6 illustrates the performance across different starting layers on various datasets. For IMDB-MULTI, we observe slightly better performance when GIP is applied from earlier layers, with a gradual decrease as the starting layer increases.

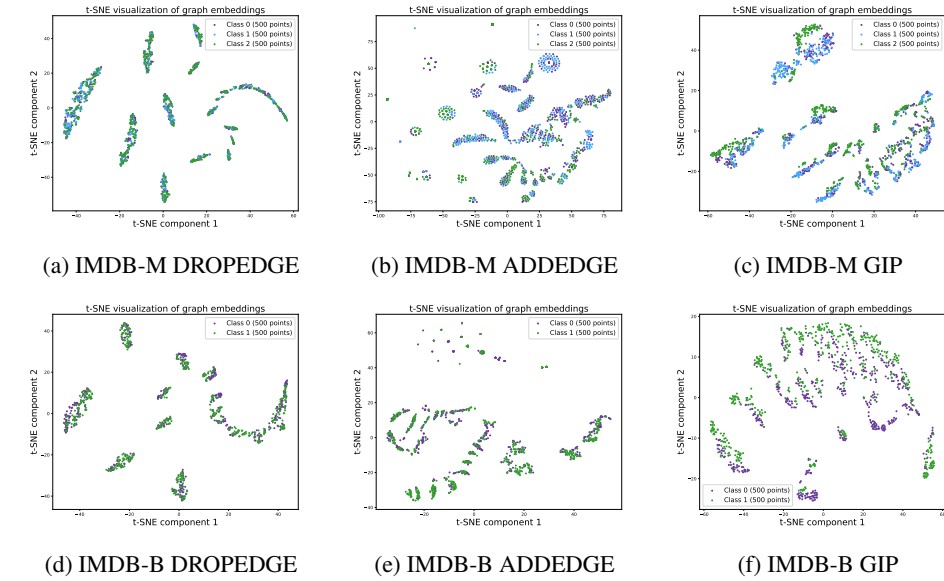

(a) IMDB-M DROPEDGE      (b) IMDB-M ADDEDGE      (c) IMDB-M GIP

(d) IMDB-B DROPEDGE      (e) IMDB-B ADDEDGE      (f) IMDB-B GIP

Figure 7: Graph representation pre-trained by GRACE w/o label. Our analysis of the t-SNE visualizations reveals that for the two most distinctive datasets, GIP significantly diminishes the overlap between different graph classes in the representation space and enhances the separation of manifolds. Furthermore, examination of the t-SNE coordinates demonstrates that it also simultaneously compresses manifold volumes.

Table 3: CMSP ↑ Scores of Different Method.

| Method | MUTAG | PROTEINS | NCI1 | IMDB-BINARY | IMDB-MULTI | DD |
|---|---|---|---|---|---|---|
| GIP | 0.6065 | 0.5544 | 0.2522 | 0.6499 | 0.4082 | 0.2676 |
| ADDEDGE | 0.5385 | 0.2838 | 0.1738 | 0.2404 | 0.2459 | 0.1953 |
| DROPEDGE | 0.5528 | 0.2568 | 0.1185 | 0.0863 | 0.1121 | 0.1768 |

In contrast, IMDB-BINARY shows remarkably stable performance across all starting layers. This stability suggests that for simpler tasks like binary classification, applying GIP at deeper layers is sufficient to achieve good performance. These results indicate that while GIP is generally robust, its optimal application point may vary depending on the complexity of the task, with more complex tasks benefiting from earlier applications of GIP.

**Effect of GIP on learned graph representations.** To visually demonstrate the effectiveness of GIP in separating graph manifolds, we employ t-SNE visualizations of pre-trained graph representations on various datasets. Figure 7 showcases the results on IMDB-M and IMDB-B datasets, which showed the largest improvements in

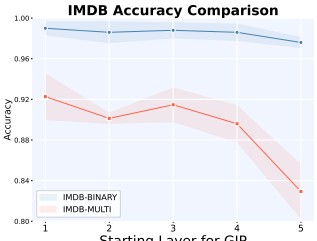

Figure 6: Effect of different starting layers of GIP

downstream tasks, similar trends are observed across other datasets, which we discussed further in Appendix E. We compare DROPEDGE, ADDEDGE, and GIP strategies on both IMDB-M (multi-class) and IMDB-B (binary) datasets. The results demonstrate GIP's superior performance in manifold separation, significantly outperforming the other two methods. For both IMDB-M and IMDB-B, GIP-generated representations exhibit clear class clustering, with points of different categories forming distinctly separated regions and only minimal overlap at boundaries. In contrast, DROPEDGE produces cluster-like structures unrelated to class labels, while ADDEDGE results in almost complete category overlap. These observations align strongly with our theoretical proof: GIP enhances mutual information between graphs within the same manifold while reducing it between graphs from different manifolds. The visualizations intuitively validate GIP's advantage in improving inter-manifold separation while preserving overall graph structural information, evident in the dispersed yet organized distribution of points.

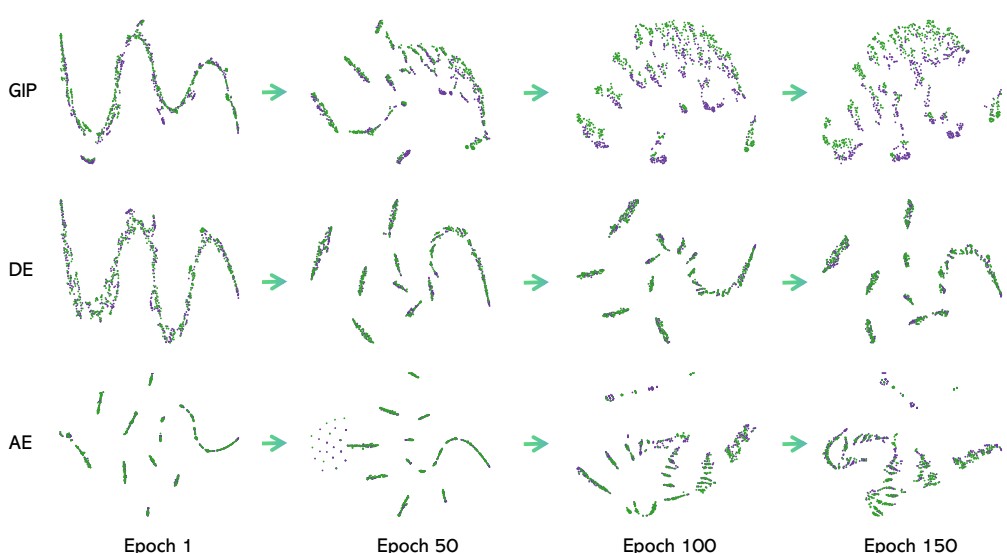

Figure 8: Evolution of graph representations during pre-training on the IMDB-BINARY dataset using the GRACE framework with three different augmentation strategies: GIP, DROPEDGE, and ADDEDGE. The t-SNE visualizations show the progression of representations at different epochs, illustrating how each strategy affects the separation of graph classes over time.

In addition to the visual representation, we define a metric called CMSP (Class-based Manifold Separation Proxy) to measure the quality of the manifold and provide numerical results in Table 3. The detailed definition and analysis are presented in Appendix F. These quantitative metrics further support our visual observations and theoretical predictions. Notably, GIP achieves excellent class separation even in the unsupervised pre-training phase. This not only supports our theoretical analysis but also highlights GIP's potential in processing complex graph data, providing a promising foundation of feature representations for downstream tasks such as graph classification.

**Evolution of Graph Representations During Pre-training.** Figure 8 illustrates the evolution of graph representations on the IMDB-BINARY dataset using GRACE, comparing GIP, DROPE-DGE, and ADDEDGE at epochs in $\{1, 50, 100, 150\}$. GIP starts with two close but distinguishable manifolds and progressively enhances their separation, achieving clear manifold bifurcation by epoch 150. DROPEDGE initially shows promise but fails to maintain manifold separation over time. ADDEDGE exhibits little manifold distinction throughout the process. This evolution demonstrates GIP's unique ability to consistently capture and enhance class-relevant features, leading to better-structured embedding manifolds. It aligns with our theoretical expectations of improved intra-manifold cohesion and inter-manifold separation, outperforming other methods in learning discriminative graph representations.

## 5 CONCLUSION

In conclusion, our work introduces Graph Interplay (GIP), a transformative approach to Graph Self-Supervised Learning (GSSL) that specifically addresses the unique challenges presented by graph-structured data. By ingeniously incorporating random inter-graph edges within batch processes, GIP capitalizes on the inherent properties of graph data, facilitating a more nuanced and effective learning process. Our theoretical and empirical analyses substantiate that GIP not only enhances the learning of graph embeddings via principled manifold separation but also significantly improves performance on downstream tasks across multiple challenging datasets. This advancement underscores the potential of tailored methodologies in fully exploiting the structural and relational complexities of graphs, paving the way for more sophisticated graph learning techniques. Moreover, GIP's compatibility with existing GNN frameworks and its computational efficiency make it a versatile and scalable solution, poised to redefine the standards of graph-based learning in self-supervised settings.

**Ethics Statement** To the authors' best knowledge, this research adheres to ethical principles and raises no ethical concerns.

**Reproducibility Statement** An anonymous link to our source code is provided in the abstract, enabling direct access to our implementation for reproduction purposes. Comprehensive information about the datasets used and implementation details are presented in Section 4.1 of the main paper and in the Appendix B.

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

Table 4: TU Benchmark Datasets and OGB chemical molecular datasets For TU datasets, the metric used for classification task is accuracy. For OGB datasets, the evaluation metric used for regression task is RMSE, and for classification is ROC-AUC.

| Data Type | Name | #Graphs | Avg. #Nodes | Avg. #Edges | #Classes/Tasks |
|---|---|---|---|---|---|
| Biochemical Molecules | NCI1 | 4,110 | 29.87 | 32.30 | 2 |
| | PROTEINS | 1,113 | 39.06 | 72.82 | 2 |
| | MUTAG | 188 | 17.93 | 19.79 | 2 |
| | DD | 1,178 | 284.32 | 715.66 | 2 |
| Social Networks | IMDB-BINARY | 1,000 | 19.8 | 96.53 | 2 |
| | IMDB-MULTI | 1,500 | 13.0 | 65.94 | 3 |
| OGB Regression | ogbg-molesol | 1,128 | 13.3 | 13.7 | 1 |
| | ogbg-molipo | 4,200 | 27.0 | 29.5 | 1 |
| | ogbg-molfreesolv | 642 | 8.7 | 8.4 | 1 |
| OGB Classification | ogbg-molbace | 1,513 | 34.1 | 36.9 | 1 |
| | ogbg-molbbbp | 2,039 | 24.1 | 26.0 | 1 |
| | ogbg-molclintox | 1,477 | 26.2 | 27.9 | 2 |

## A  MORE RELATED WORKS

**Graph Neural Networks.** Graph Neural Networks (GNNs) have become fundamental in processing graph-structured data, showing success across various domains. From the initial concept introduced by Scarselli et al. (2008) to more advanced models like GCNs (Kipf & Welling, 2016a), GraphSAGE (Hamilton et al., 2017), and GAT (Veličković et al., 2017), GNNs have evolved to handle complex graph structures efficiently. The Message Passing Neural Network (MPNN) framework (Gilmer et al., 2017) unified various GNN architectures, highlighting commonalities in message-passing operations. Efforts to enhance GNN expressiveness and depth, such as GIN (Xu et al., 2018) and DeepGCNs (Li et al., 2019), have further expanded their capabilities. Techniques like DropEdge (Rong et al., 2019) and PairNorm (Zhao & Akoglu, 2019) mitigate challenges in training deep GNNs, particularly the over-smoothing problem. Comprehensive surveys by Wu et al. (2020), Zhou et al. (2020), and Khoshraftar & An (2024) provide detailed overviews of GNN advancements and applications.

## B  IMPLEMENTATION DETAILS

**Training configuration.** For each framework, we implement it based on (Zhu et al., 2021a) [1]. We used the following hyperparameters: a learning rate of $5 \times 10^{-4}$, a node hidden size of $512$, and a varying number of GCN encoder layers selected from $\{2, 3, 4, 5\}$. For all graph classification datasets, the number of training epochs was chosen from $\{20, 40, \ldots, 200\}$. To achieve performance closer to the global optimum, we conducted 20 randomized searches to determine the optimal parameters for edge perturbation. For each parameter configuration, performance was evaluated using 5 different random seeds, from which the mean and standard deviation were computed. The best-performing parameter configuration among the 20 searches was then selected, and the corresponding results were reported. For all graph classification datasets, the batch size was set to $\{32, 64, 128\}$. We use exactly the same setup to search for the optimal edge perturbation probability to evaluate DROPEDGE and ADDEDGE.

**Datasets.** The TU dataset is a classic graph classification benchmark, where graph objects include mutagenic compounds, chemical compounds, protein structures, ego networks based on movie partnerships, and more. While the OGBG dataset we use focuses on molecular property prediction, such as some Physical Chemistry and Physiology properties. Compared to the TU dataset, OGBG graphs are relatively sparse with limited topological patterns due to similar numbers of nodes and edges.

---

[1] https://github.com/PyGCL/PyGCL

## C GSSL OBJECTIVE FUNCTION

This section presents the loss functions of four representative graph self-supervised learning methods for graph-level tasks: GRACE, MVGRL, BGRL, and G-BT. These methods can be categorized into two main approaches: mutual information maximization and redundancy reduction. GIP is implemented within all four frameworks.

GRACE and MVGRL both aim to maximize mutual information using different estimators. GRACE utilizes an InfoNCE estimator for graph-level representations:

$$\mathcal{L}_{GRACE} = -\log \frac{\exp(s(f_g(\mathcal{G}_i), f_g(\mathcal{G}_i'))/\tau)}{\sum_{j=1}^{N} \exp(s(f_g(\mathcal{G}_i), f_g(\mathcal{G}_j'))/\tau)} \tag{7}$$

where $f_g(\mathcal{G}_i)$ and $f_g(\mathcal{G}_i')$ are graph embeddings of two views of the same graph, $s(\cdot, \cdot)$ is a similarity function, and $\tau$ is a temperature parameter.

MVGRL employs the Jensen-Shannon MI estimator to maximize mutual information between different structural views of graphs:

$$\mathcal{L}_{MVGRL} = \hat{I}^{(JS)}(f_g(\mathcal{G}), f_g(\mathcal{G}')) \tag{8}$$

where $f_g(\mathcal{G})$ and $f_g(\mathcal{G}')$ are graph-level representations from two different views, and $\hat{I}^{(JS)}$ is the Jensen-Shannon MI estimator defined as:

$$\hat{I}^{(JS)}(f_g(\mathcal{G}), f_g(\mathcal{G}')) = \mathbb{E}_{(\mathcal{G},\mathcal{G}')\sim\mathcal{P}}[\log(\mathcal{D}(f_g(\mathcal{G}), f_g(\mathcal{G}')))] + \mathbb{E}_{(\mathcal{G},\mathcal{G}')\sim\mathcal{P}\times\mathcal{P}}[\log(1-\mathcal{D}(f_g(\mathcal{G}), f_g(\mathcal{G}')))] \tag{9}$$

Here, $\mathcal{D}$ is a discriminator function, and $\mathcal{P}$ represents the distribution of graph pairs.

In contrast, BGRL and G-BT adopt the redundancy reduction principle. BGRL's loss function is inspired by BYOL and implicitly reduces redundancy through its bootstrapping mechanism:

$$\mathcal{L}_{BGRL} = \|sg(f_t(\mathcal{G}')) - f_o(\mathcal{G})\|^2 \tag{10}$$

where $f_t$ and $f_o$ are the target and online networks respectively, $\mathcal{G}$ and $\mathcal{G}'$ are two augmented views of a graph, and $sg$ denotes stop-gradient.

G-BT explicitly employs a redundancy reduction objective:

$$\mathcal{L}_{G-BT} = \underbrace{\sum_i (1 - C_{ii})^2}_{\text{invariance term}} + \lambda \underbrace{\sum_i \sum_{j\neq i} C_{ij}^2}_{\text{redundancy reduction term}} \tag{11}$$

where $C$ is the cross-correlation matrix between embeddings of different views, and $\lambda$ is a trade-off parameter.

## D EFFECT OF TWO-BRANCH DROPEDGE/ADDEDGE PARAMETERS

In this section, we present a detailed analysis of the ADDEDGE and DROPEDGE methods, comparing their performance across various datasets from the TU Dataset collection. As a supplement to Figure 4 in the main body, we analyze the GRACE framework as a case study here. Figures 9b and 9a visualize the results as 3D surface plots, where the $x$ and $y$ axes represent the probabilities of adding or dropping edges, respectively, and the $z$-axis represents the achieved accuracy.

The DROPEDGE method, as shown in Figure 9a, exhibits complex and highly dataset-dependent behavior. Across the six datasets (MUTAG, IMDB-MULTI, IMDB-BINARY, PROTEINS, NCI1, and DD), we observe no consistent optimal probability for edge dropping. Instead, each dataset presents a unique surface with varying patterns of peaks and valleys. For instance, MUTAG shows the highest accuracy when both dropping probabilities are low, while DD exhibits a distinctive pattern where accuracy peaks when one probability is high and the other is low. This variability suggests that the effectiveness of DROPEDGE is strongly influenced by the specific structural characteristics of each dataset. Similarly, the ADDEDGE method, visualized in Figure 9b, demonstrates equally

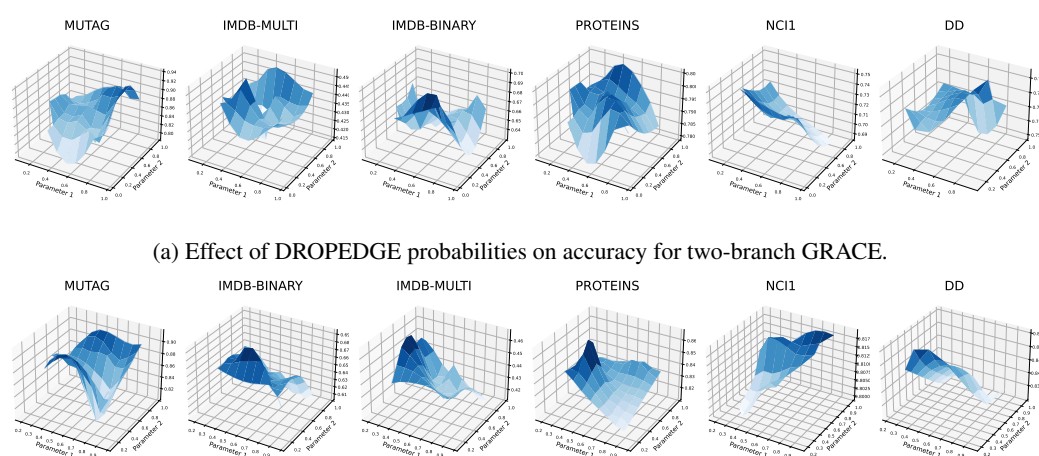

(a) Effect of DROPEDGE probabilities on accuracy for two-branch GRACE.

(b) Effect of ADDEDGE probabilities on accuracy for two-branch GRACE.

Figure 9: Parameter sensitivity analysis for two-branch GRACE with DROPEDGE and ADDEDGE

complex and dataset-specific performance patterns. While some datasets like NCI1 show improved performance at higher edge addition probabilities, others like DD achieve the best results at lower probabilities. The IMDB datasets (BINARY and MULTI) present particularly intricate surfaces with multiple local optima, highlighting the challenge of finding optimal parameters for these methods.

When compared to GIP, both ADDEDGE and DROPEDGE lack a consistent trend of improvement with increasing probabilities that GIP exhibits. This inconsistency makes these methods potentially more challenging to tune and less reliable across different datasets. However, the complex surfaces observed for ADDEDGE and DROPEDGE suggest that these methods might capture more nuanced structural information, albeit at the cost of increased sensitivity to parameter settings. We conducted the same experiment within the BGRL framework and found consistent patterns, as shown in Figure 10.

In conclusion, while ADDEDGE and DROPEDGE show potential for performance improvements in specific scenarios, their highly variable behavior across datasets makes them less reliable compared to the more consistent GIP method. These findings not only validate the effectiveness of GIP but also highlight the complex relationship between graph structure manipulation and representation quality. The dataset-specific optimalities observed in ADDEDGE and DROPEDGE suggest that there might be untapped potential in more fine-grained graph manipulation strategies. Future research could focus on developing more sophisticated versions of GIP that adaptively adjust edge addition strategies based on specific graph properties or dataset characteristics. This could involve incorporating graph structural features, node attributes, or even learned representations to guide the inter-graph edge addition process.

# E  ANALYSIS OF THE QUALITY OF THE LEARNED REPRESENTATION

In this section, we present 2D and 3D visualizations of graph representations pre-trained by GRACE with and without our GIP method. Figure 11 shows t-SNE projections of graph embeddings for three datasets: NCI1, PROTEINS, and DD. For each dataset, we compare three scenarios: DROPEDGE, ADDEDGE, and GIP.

Taking the NCI1 dataset as an example (subfigures a, b, and c), we observe a high degree of overlap between data points from different manifolds (classes) in the DROPEDGE and ADDEDGE-derived representation distributions. In contrast, GIP significantly reduces this inter-manifold overlap. Although GIP does not produce two entirely separate clusters in the representation space, it is evident that the distributions of the two manifolds have been shifted relative to each other, resulting in improved separation.

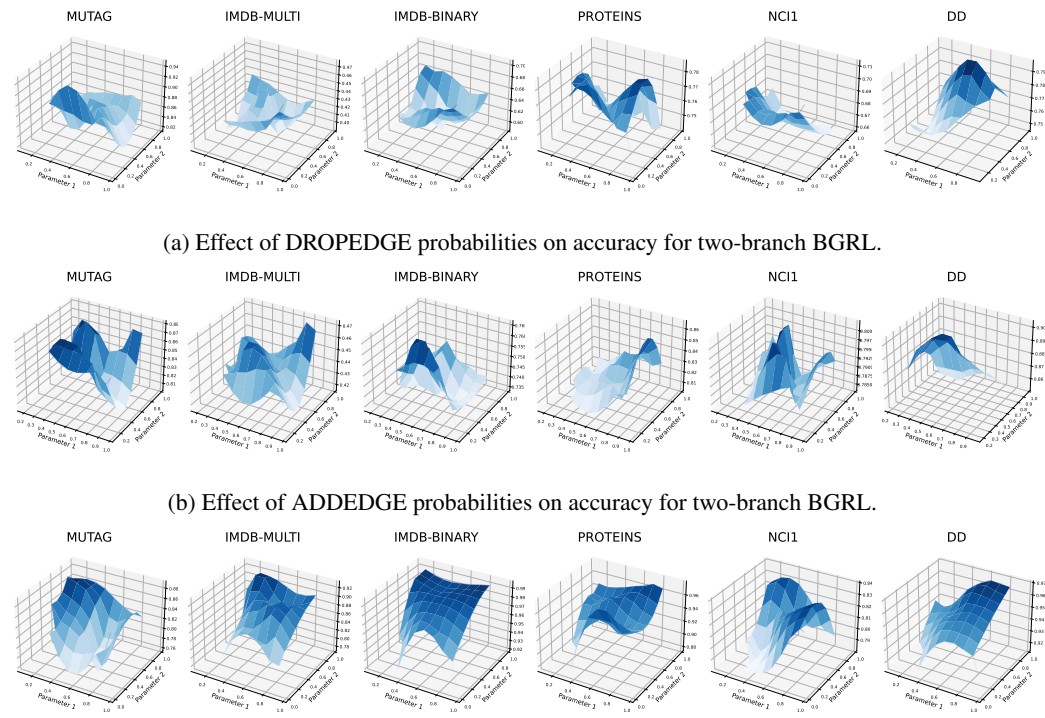

(a) Effect of DROPEDGE probabilities on accuracy for two-branch BGRL.

(b) Effect of ADDEDGE probabilities on accuracy for two-branch BGRL.

(c) Effect of GIP probabilities on accuracy for two-branch BGRL.

Figure 10: Parameter sensitivity analysis for BGRL with different methods.

This reduction in manifold overlap is crucial for downstream tasks. The overlap of data points from different manifolds can be detrimental, as it directly leads to indistinguishable initial features, making classification more challenging. GIP's ability to enhance manifold separation suggests that it produces more discriminative features, which can significantly benefit downstream tasks.

Similar trends of improved manifold separation can be observed in the PROTEINS (subfigures d, e, and f) and DD (subfigures g, h, and i) datasets. In both cases, GIP consistently shows clearer boundaries between manifolds compared to DROPEDGE and ADDEDGE. These visual results provide intuitive support for our theoretical analysis, demonstrating that GIP indeed enhances the separation between different manifolds in the embedding space. This improved manifold separation likely contributes to the enhanced performance of GIP in downstream tasks, as it allows for more discriminative graph representations that better reflect the underlying manifold structure of the data.

## F    CLASS-BASED MANIFOLD SEPARATION PROXY (CMSP)

To quantitatively evaluate the effectiveness of graph embedding methods in preserving and potentially enhancing the underlying manifold structure, we introduce the Class-based Manifold Separation Proxy (CMSP). This metric is designed to assess how well the embedding method distinguishes between different classes of graphs in the embedded space, serving as a proxy for manifold separation. We base this approach on the assumption that graphs from the same class are likely to lie on or near the same manifold in the high-dimensional space, while graphs from different classes are likely to lie on different manifolds. While we do not have direct access to the true manifold structure, we use class labels as proxies for manifold assignments. This allows us to quantify the degree of separation between these assumed manifolds in the embedding space. The CMSP is particularly relevant for supervised learning tasks such as graph classification, where the goal is to distinguish between different classes of graphs. The CMSP is defined through a series of calculations on the embedded representations. First, we compute the Intra-class Dispersion ($D_k$) for each class $k$, which

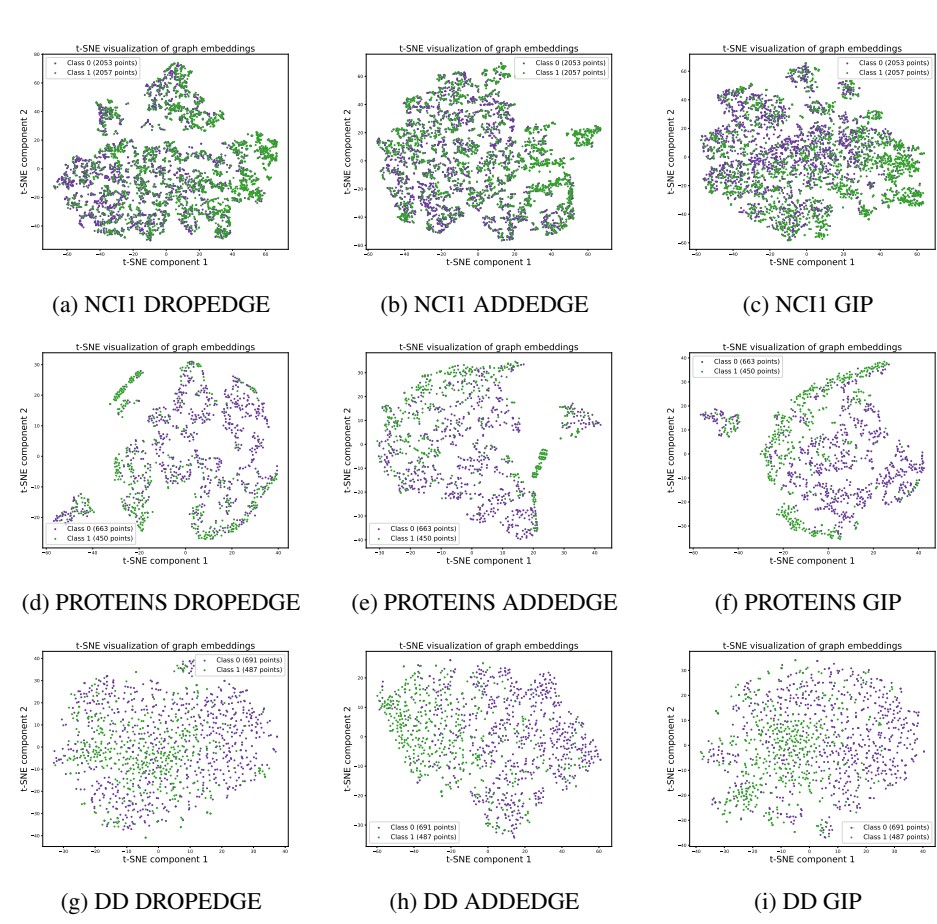

(a) NCI1 DROPEDGE     (b) NCI1 ADDEDGE     (c) NCI1 GIP

(d) PROTEINS DROPEDGE    (e) PROTEINS ADDEDGE    (f) PROTEINS GIP

(g) DD DROPEDGE     (h) DD ADDEDGE     (i) DD GIP

Figure 11: t-SNE visualizations of graph embeddings pre-trained by GRACE with DROPEDGE, ADDEDGE, and GIP on NCI1, PROTEINS, and DD datasets. Each row represents a dataset, and each column represents a different method. GIP (rightmost column) mitigates the severe overlap of data points from different classes observed in DROPEDGE and ADDEDGE (left and middle columns). This improved separation between manifolds in the embedding space suggests that GIP produces more discriminative features, potentially benefiting downstream tasks.

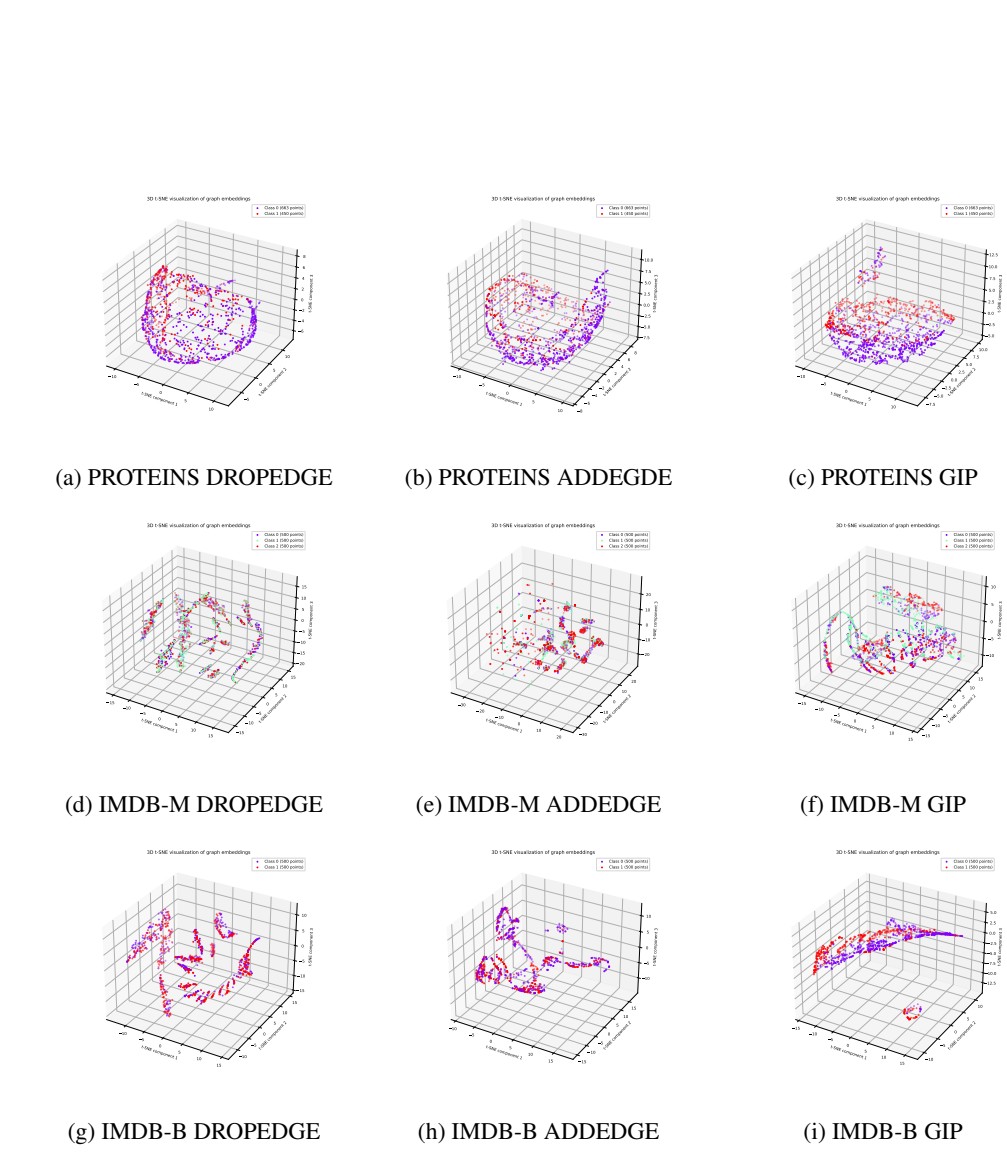

(a) PROTEINS DROPEDGE     (b) PROTEINS ADDEGDE     (c) PROTEINS GIP

(d) IMDB-M DROPEDGE     (e) IMDB-M ADDEDGE     (f) IMDB-M GIP

(g) IMDB-B DROPEDGE     (h) IMDB-B ADDEDGE     (i) IMDB-B GIP

Figure 12: 3D T-SNE visualizations of graph embeddings for PROTEINS, IMDB-MULTI (IMDB-M), IMDB-BINARY (IMDB-B), and NCI1 datasets. Each row represents a different dataset, while columns show results for DROPEDGE, ADDEDGE, and GIP methods respectively. Colors represent different classes within each dataset. Notable observations include: (a-c) For PROTEINS, GIP achieves better class separation compared to DROPEDGE and ADDEDGE. (d-f) IMDB-MULTI shows a more structured distribution with GIP, though class overlap remains. (g-i) In IMDB-BINARY, GIP produces a more distinct separation between classes, forming a clearer boundary.

we interpret as the dispersion within a manifold:

$$D_k = \frac{1}{n_k^2} \sum_{i \neq j} \|x_i^k - x_j^k\| \tag{12}$$

where $x_i^k$ is the embedding vector of the $i$-th sample in class $k$, and $n_k$ is the number of samples in class $k$. We then calculate the Average Intra-class Dispersion ($D_{avg}$) across all $K$ classes:

$$D_{avg} = \frac{1}{K} \sum_{k=1}^{K} D_k \tag{13}$$

To measure the separation between classes, which we interpret as separation between manifolds, we compute the Inter-class Separation ($S$) as the average distance between class centroids:

$$S = \frac{2}{K(K-1)} \sum_{i<j} \|\mu_i - \mu_j\| \tag{14}$$

where $\mu_k = \frac{1}{n_k} \sum_{i=1}^{n_k} x_i^k$ is the centroid of class $k$. Finally, we define the Class-based Manifold Separation Proxy (CMSP) as the ratio of inter-class separation to intra-class dispersion:

$$CMSP = \frac{S}{D_{avg}} \tag{15}$$

A higher CMSP value indicates better separation between classes in the embedding space, which we interpret as improved separation between the underlying manifolds. This metric allows for a direct comparison between different embedding methods, capturing their ability to produce representations that preserve and potentially enhance the manifold structure of the data, as approximated by class labels. It's important to note that while we use class labels as proxies for manifold assignments, this approach has limitations. The true manifold structure of the data may be more complex than what is captured by class labels alone. However, in the context of graph classification tasks, where the goal is often to distinguish between different classes of graphs, this approximation provides a practical and interpretable measure of embedding quality and manifold separation.

## G  ENHANCED MANIFOLD SEPARATION IN GRAPH INTERPLAY (GIP)

### G.1  DEFINITIONS AND ASSUMPTIONS

**Definition 1** (Graph Set and Intrinsic Manifolds). *Let $\mathcal{S} = \{\mathcal{G}_1, \mathcal{G}_2, \ldots, \mathcal{G}_N\}$ be a set of $N$ graphs. Assume these graphs lie on $K$ underlying manifolds $\mathcal{F} = \{\mathcal{M}_1, \mathcal{M}_2, \ldots, \mathcal{M}_K\}$ in a high-dimensional space. Define the mapping function $\mu : \mathcal{G} \rightarrow \{1, \ldots, K\}$ that assigns each graph to its corresponding manifold.*

**Definition 2** (Graph Distribution). *For each manifold $\mathcal{M}_k$, assume there exists a probability distribution $P_k$ from which graphs on $\mathcal{M}_k$ are sampled. Let $\mathcal{G} \sim P_k$ denote a graph randomly sampled from manifold $\mathcal{M}_k$.*

**Definition 3** (SSL Embedding). *Let $f_s : \mathcal{G} \rightarrow \mathbb{R}^d$ be the well-trained GNN embedding function obtained through SSL, which maps graphs to a $d$-dimensional Euclidean space.*

**Definition 4** (Manifold-Relevant Information). *For a manifold $\mathcal{M}_k$, we define the manifold-relevant information $Z_k$ as a random variable representing the embedding of a graph randomly sampled from $\mathcal{M}_k$:*

$$Z_k = f_s(\mathcal{G}), \quad \mathcal{G} \sim P_k \tag{16}$$

*where $P_k$ is the probability distribution over graphs in manifold $\mathcal{M}_k$, and $f_s$ is the SSL embedding function.*

**Lemma 1** (GIP Transformation). *Consider a GNN with $n$ layers ($n \geq 1$) used in Graph Interplay (GIP), under the following conditions:*

- *Each layer of the GNN consists of a linear transformation followed by a ReLU activation function.*

- *The pooling operation used to obtain graph-level representations is additive.*

*The GIP transformation can be equivalently represented as:*

$$f_g(\mathcal{G}_i) = f(\mathcal{G}_i) + \sum_{j \neq i} \alpha_{ij} f(\mathcal{G}_j) \tag{17}$$

*where $f : \mathcal{G} \to \mathbb{R}^d$ is a GNN encoder, and $\alpha_{ij}$ are learnable parameters representing the strength of interaction between graphs $\mathcal{G}_i$ and $\mathcal{G}_j$.*

*Proof.* We prove this by induction on the number of layers $n$.

Base case ($n = 1$): Let $\mathcal{G}_i = (V_i, E_i)$ be a graph in the batch, and $\mathcal{G}_i^{GIP}$ be the augmented graph after GIP's inter-graph edge additions.

For a node $v \in V_i$, its representation after one layer of GNN on $\mathcal{G}_i^{GIP}$ is:

$$h_v^{(1)} = \text{ReLU}(W^{(1)} \sum_{u \in N(v)} x_u + b^{(1)}) \tag{18}$$

where $N(v)$ is the neighborhood of $v$ in $\mathcal{G}_i^{GIP}$, $x_u$ is the input feature of node $u$, $W^{(1)}$ is the weight matrix, and $b^{(1)}$ is the bias term.

We can separate this sum into contributions from $\mathcal{G}_i$ and other graphs:

$$h_v^{(1)} = \text{ReLU}(W^{(1)}(\sum_{u \in N(v) \cap V_i} x_u + \sum_{j \neq i} \sum_{u \in N(v) \cap V_j} x_u) + b^{(1)}) \tag{19}$$

Define $y_v^{(1)} = W^{(1)} \sum_{u \in N(v) \cap V_i} x_u + b^{(1)}$ and $z_v^{(1)} = W^{(1)} \sum_{j \neq i} \sum_{u \in N(v) \cap V_j} x_u$. Then:

$$h_v^{(1)} = \text{ReLU}(y_v^{(1)} + z_v^{(1)}) = \text{ReLU}(y_v^{(1)}) + \text{ReLU}(y_v^{(1)} + z_v^{(1)}) - \text{ReLU}(y_v^{(1)}) \tag{20}$$

The graph-level representation is obtained by additive pooling:

$$f_g^{(1)}(\mathcal{G}_i) = \sum_{v \in V_i} h_v^{(1)} = \sum_{v \in V_i} \text{ReLU}(y_v^{(1)}) + \sum_{v \in V_i} [\text{ReLU}(y_v^{(1)} + z_v^{(1)}) - \text{ReLU}(y_v^{(1)})] \tag{21}$$

The first term is $f^{(1)}(\mathcal{G}_i)$, and we can define:

$$\alpha_{ij}^{(1)} = \frac{\sum_{v \in V_i} [\text{ReLU}(y_v^{(1)} + z_v^{(1)}) - \text{ReLU}(y_v^{(1)})]}{f^{(1)}(\mathcal{G}_j)} \tag{22}$$

Thus, $f_g^{(1)}(\mathcal{G}_i) = f^{(1)}(\mathcal{G}_i) + \sum_{j \neq i} \alpha_{ij}^{(1)} f^{(1)}(\mathcal{G}_j)$ holds for $n = 1$.

Inductive step: Assume the lemma holds for $n = k$ layers. We prove it holds for $n = k + 1$ layers.

For the $(k + 1)$-th layer, the representation of a node $v$ is:

$$h_v^{(k+1)} = \text{ReLU}(W^{(k+1)} \sum_{u \in N(v)} h_u^{(k)} + b^{(k+1)}) \tag{23}$$

By the induction hypothesis:

$$h_u^{(k)} = h_u^{(k)}(\mathcal{G}_i) + \sum_{j \neq i} \beta_{ij}^{(k)} h_u^{(k)}(\mathcal{G}_j) \tag{24}$$

Substituting this into the $(k + 1)$-th layer equation:

$$h_v^{(k+1)} = \text{ReLU}(W^{(k+1)}(\sum_{u \in N(v)} h_u^{(k)}(\mathcal{G}_i) + \sum_{j \neq i} \sum_{u \in N(v)} \beta_{ij}^{(k)} h_u^{(k)}(\mathcal{G}_j)) + b^{(k+1)}) \tag{25}$$

Define:

$$y_v^{(k+1)} = W^{(k+1)} \sum_{u \in N(v)} h_u^{(k)}(\mathcal{G}_i) + b^{(k+1)} \tag{26}$$

$$z_v^{(k+1)} = W^{(k+1)} \sum_{j \neq i} \sum_{u \in N(v)} \beta_{ij}^{(k)} h_u^{(k)}(\mathcal{G}_j) \tag{27}$$

Following the same steps as in the base case:

$$f_g^{(k+1)}(\mathcal{G}_i) = f^{(k+1)}(\mathcal{G}_i) + \sum_{j \neq i} \alpha_{ij}^{(k+1)} f^{(k+1)}(\mathcal{G}_j) \tag{28}$$

where

$$\alpha_{ij}^{(k+1)} = \frac{\sum_{v \in V_i} [\mathrm{ReLU}(y_v^{(k+1)} + z_v^{(k+1)}) - \mathrm{ReLU}(y_v^{(k+1)})]}{f^{(k+1)}(\mathcal{G}_j)} \tag{29}$$

By induction, the lemma holds for any number of layers $n \geq 1$. $\square$

**Assumption 1** (Expected Intra-Manifold Information Consistency for SSL). *For each manifold $\mathcal{M}_k$, the SSL embedding function $f_s$ satisfies:*

$$\mathbb{E}_{\mathcal{G}_i \sim P_k}[I(f_s(\mathcal{G}_i); Z_k)] > \mathbb{E}_{\mathcal{G}_i \sim P_k}[\max_{l \neq k} I(f_s(\mathcal{G}_i); Z_l)] \tag{30}$$

*where $I(\cdot; \cdot)$ denotes mutual information, and the expectation is taken over graphs $\mathcal{G}_i$ sampled from the distribution $P_k$ of manifold $\mathcal{M}_k$.*

**Assumption 2** (Self-Supervised Learning Objective). *The self-supervised learning objective for GIP is approximated in terms of mutual information as:*

$$\mathcal{L} = \mathbb{E}_{\mathcal{G}_i} \left[ -I(f_g^{(1)}(\mathcal{G}_i); f_g^{(2)}(\mathcal{G}_i)) + \lambda \mathbb{E}_{\mathcal{G}_j \neq \mathcal{G}_i}[I(f_g^{(1)}(\mathcal{G}_i); f_g^{(1)}(\mathcal{G}_j))] \right] \tag{31}$$

*where $f_g^{(1)}$ and $f_g^{(2)}$ represent two different views of $\mathcal{G}_i$, $\lambda > 0$ is a balancing parameter, and the expectations are taken over all graphs in the dataset.*

### G.2 MAIN THEOREM AND PROOF

**Theorem 1** (GIP's Improvement on Manifold Separation). *Given the above definitions and assumptions, under the self-supervised learning objective and sufficient training, GIP can achieve better expected manifold separation than SSL:*

$$\frac{\mathbb{E}_{\mathcal{G}_i \sim P_k}[I(f_g^{(v)}(\mathcal{G}_i); Z_k)]}{\max_{l \neq k} \mathbb{E}_{\mathcal{G}_i \sim P_k}[I(f_g^{(v)}(\mathcal{G}_i); Z_l)]} > \frac{\mathbb{E}_{\mathcal{G}_i \sim P_k}[I(f_s(\mathcal{G}_i); Z_k)]}{\max_{l \neq k} \mathbb{E}_{\mathcal{G}_i \sim P_k}[I(f_s(\mathcal{G}_i); Z_l)]}, \quad v \in \{1, 2\} \tag{32}$$

*where $I(\cdot; \cdot)$ denotes mutual information and $f_g^{(v)}$ represents the GIP embedding function for view $v$.*

*Proof.* Note that throughout this proof, $f_s$ denotes the GNN that has been well-trained through standard SSL, serving as our baseline, while $f_g^{(v)}$ represents the GIP embedding function built upon $f_s$. Our proof consists of two main steps:

- Step 1: We show that optimizing the contrastive learning objective leads GIP to learn coefficients that approach the optimal configuration for manifold separation.

- Step 2: We demonstrate that with these optimized coefficients, GIP achieves better manifold separation than the original SSL embedding.

Step 1: Convergence to Optimal Coefficients

Let's expand the self-supervised learning objective using the definition of GIP transformation:

$$\mathcal{L} = \mathbb{E}_{\mathcal{G}_i} \left[ -I(f_s(\mathcal{G}_i) + \sum_{k \neq i} \alpha_{ik}^{(1)} f_s(\mathcal{G}_k); f_s(\mathcal{G}_i) + \sum_{k \neq i} \alpha_{ik}^{(2)} f_s(\mathcal{G}_k)) \right. \tag{33}$$

$$\left. + \lambda \mathbb{E}_{\mathcal{G}_j \neq \mathcal{G}_i}[I(f_s(\mathcal{G}_i) + \sum_{k \neq i} \alpha_{ik}^{(1)} f_s(\mathcal{G}_k); f_s(\mathcal{G}_j) + \sum_{k \neq j} \alpha_{jk}^{(1)} f_s(\mathcal{G}_k))] \right] \tag{34}$$

From the Expected Intra-Manifold Information Consistency assumption, we know that for each manifold $\mathcal{M}_k$:

$$\mathbb{E}_{\mathcal{G}_i \sim P_k}[I(f_s(\mathcal{G}_i); Z_k)] > \mathbb{E}_{\mathcal{G}_i \sim P_k}[\max_{l \neq k} I(f_s(\mathcal{G}_i); Z_l)] \tag{35}$$

This implies that for $\mathcal{G}_i, \mathcal{G}_j \in \mathcal{M}_k$:

$$\mathbb{E}_{\mathcal{G}_i, \mathcal{G}_j \sim P_k}[I(f_s(\mathcal{G}_i); f_s(\mathcal{G}_j))] > \mathbb{E}_{\mathcal{G}_i \sim P_k, \mathcal{G}_j \sim P_l, l \neq k}[I(f_s(\mathcal{G}_i); f_s(\mathcal{G}_j))] \tag{36}$$

Given this property, the gradient of $\mathcal{L}$ with respect to $\alpha_{ij}^{(v)}$ behaves as follows:

$$\mathbb{E}_{\mathcal{G}_i, \mathcal{G}_j} \left[ \frac{\partial \mathcal{L}}{\partial \alpha_{ij}^{(v)}} \right] = \begin{cases} < 0, & \text{if } \mu(\mathcal{G}_i) = \mu(\mathcal{G}_j) \\ > 0, & \text{if } \mu(\mathcal{G}_i) \neq \mu(\mathcal{G}_j) \end{cases} \tag{37}$$

This gradient behavior is a direct consequence of the Expected Intra-Manifold Information Consistency. When $\mathcal{G}_i$ and $\mathcal{G}_j$ are from the same manifold, increasing $\alpha_{ij}^{(v)}$ will increase the mutual information in the first term of $\mathcal{L}$ more than it increases the second term, resulting in a negative gradient. Conversely, when $\mathcal{G}_i$ and $\mathcal{G}_j$ are from different manifolds, increasing $\alpha_{ij}^{(v)}$ will increase the second term more than the first, resulting in a positive gradient.

Based on this gradient behavior, we define the optimal coefficient configuration $\alpha_{ij}^{opt}$ as:

$$\alpha_{ij}^{opt} = \begin{cases} > 0, & \text{if } \mu(\mathcal{G}_i) = \mu(\mathcal{G}_j) \\ 0, & \text{if } \mu(\mathcal{G}_i) \neq \mu(\mathcal{G}_j) \end{cases} \tag{38}$$

While the actual learned coefficients may not achieve this exact configuration due to finite training time and the stochastic nature of optimization, we can show that the GIP transformation with these optimal coefficients provides an upper bound on the manifold separation capability of GIP.

Step 2: Improved Manifold Separation

Given the optimal coefficients $\alpha_{ij}^{opt}$, for $\mathcal{G}_i \in \mathcal{M}_k$, we have:

$$f_g^{opt}(\mathcal{G}_i) = f_s(\mathcal{G}_i) + \sum_{j:\mu(\mathcal{G}_j)=k, j \neq i} \alpha_{ij}^{opt} f_s(\mathcal{G}_j) \tag{39}$$

Now, let's analyze the mutual information:

$$\mathbb{E}_{\mathcal{G}_i \sim P_k}[I(f_g^{opt}(\mathcal{G}_i); Z_k)] = \mathbb{E}_{\mathcal{G}_i \sim P_k}[I(f_s(\mathcal{G}_i) + \sum_{j:\mu(\mathcal{G}_j)=k, j \neq i} \alpha_{ij}^{opt} f_s(\mathcal{G}_j); Z_k)] \tag{40}$$

$$> \mathbb{E}_{\mathcal{G}_i \sim P_k}[I(f_s(\mathcal{G}_i); Z_k)] \tag{41}$$

The strict inequality holds because we are adding strictly positive weighted information from the same manifold, which increases the mutual information with $Z_k$.

For $l \neq k$:

$$\mathbb{E}_{\mathcal{G}_i \sim P_k}[I(f_g^{opt}(\mathcal{G}_i); Z_l)] = \mathbb{E}_{\mathcal{G}_i \sim P_k}[I(f_s(\mathcal{G}_i) + \sum_{j:\mu(\mathcal{G}_j)=k, j \neq i} \alpha_{ij}^{opt} f_s(\mathcal{G}_j); Z_l)] \tag{42}$$

$$= \mathbb{E}_{\mathcal{G}_i \sim P_k}[I(f_s(\mathcal{G}_i); Z_l)] \tag{43}$$

The equality holds because, on average, the additional information from $\mathcal{M}_k$ is expected to provide no new information about $Z_l$ beyond what is already contained in $f_s(\mathcal{G}_i)$.

Combining these results:

$$\frac{\mathbb{E}_{\mathcal{G}_i \sim P_k}[I(f_g^{opt}(\mathcal{G}_i); Z_k)]}{\max_{l \neq k} \mathbb{E}_{\mathcal{G}_i \sim P_k}[I(f_g^{opt}(\mathcal{G}_i); Z_l)]} > \frac{\mathbb{E}_{\mathcal{G}_i \sim P_k}[I(f_s(\mathcal{G}_i); Z_k)]}{\max_{l \neq k} \mathbb{E}_{\mathcal{G}_i \sim P_k}[I(f_s(\mathcal{G}_i); Z_l)]} \tag{44}$$

Since $f_g^{opt}$ represents the ideal case for GIP, we expect the actual GIP transformation $f_g^{(v)}$ to approach this performance as training progresses. More precisely, for any $\epsilon > 0$ and $\delta > 0$, we conjecture that there exists a sufficiently large number of training steps $T$, such that for $t > T$:

$$P\left(\left|\frac{\mathbb{E}_{\mathcal{G}_i \sim P_k}[I(f_g^{(v)}(\mathcal{G}_i, t); Z_k)]}{\max_{l \neq k} \mathbb{E}_{\mathcal{G}_i \sim P_k}[I(f_g^{(v)}(\mathcal{G}_i, t); Z_l)]} - \frac{\mathbb{E}_{\mathcal{G}_i \sim P_k}[I(f_g^{opt}(\mathcal{G}_i); Z_k)]}{\max_{l \neq k} \mathbb{E}_{\mathcal{G}_i \sim P_k}[I(f_g^{opt}(\mathcal{G}_i); Z_l)]}\right| < \epsilon\right) > 1 - \delta \tag{45}$$

It's important to note that this convergence holds for both views $v \in \{1, 2\}$. The reason both views converge to similar performance lies in the structure of the contrastive learning objective:

$$\mathcal{L} = \mathbb{E}_{\mathcal{G}_i}\left[-I(f_g^{(1)}(\mathcal{G}_i); f_g^{(2)}(\mathcal{G}_i)) + \lambda \mathbb{E}_{\mathcal{G}_j \neq \mathcal{G}_i}[I(f_g^{(1)}(\mathcal{G}_i); f_g^{(1)}(\mathcal{G}_j))]\right] \tag{46}$$

The first term $-I(f_g^{(1)}(\mathcal{G}_i); f_g^{(2)}(\mathcal{G}_i))$ encourages agreement between the two views. As this term is minimized, the representations produced by $f_g^{(1)}$ and $f_g^{(2)}$ become increasingly similar. Simultaneously, the second term encourages both views to learn representations that separate different graphs, particularly those from different manifolds.

As a result, both views are driven to learn similar coefficient configurations that optimize the trade-off between intra-graph consistency (across views) and inter-graph discrimination. This leads to both views converging to representations that are not only similar to each other but also approach the optimal manifold separation capability represented by $f_g^{opt}$.

This completes the proof, demonstrating that GIP achieves better expected manifold separation than the original SSL embedding for both views. $\square$

**Discussion:** While our theoretical analysis demonstrates that GIP improves manifold separation by increasing intra-manifold mutual information while keeping inter-manifold mutual information constant, it's important to note that this represents a conservative lower bound on GIP's potential. In practice, GIP is likely to achieve even better separation for two reasons:

- Joint Optimization: Our analysis assumes that GIP operates on a fixed representation space learned by standard SSL. However, GIP trains the entire model from scratch, allowing for joint optimization of the base representation and the inter-graph attention mechanism. This joint optimization process is analogous to the Expectation-Maximization (EM) algorithm, where the model iteratively refines both the learned representations and the manifold structure.

- Non-linear Transformations: Our analysis considers only linear combinations of SSL-learned representations. In practice, GIP employs non-linear transformations through its neural network architecture, potentially allowing for more complex and effective manifold separations.

### G.3 EXTENSION TO BARLOW TWINS LOSS

While our main theoretical analysis focuses on the objective of maximizing mutual information, the principles of GIP can be extended to other self-supervised learning frameworks, such as the Barlow Twins (BT) loss. Adapted for graph-level representations in GIP, the BT loss can be expressed as:

$$\mathcal{L}_{G-BT} = \underbrace{\sum_i (1 - C_{ii})^2}_{\text{invariance term}} + \lambda \underbrace{\sum_i \sum_{j \neq i} C_{ij}^2}_{\text{redundancy reduction term}} \tag{47}$$

where $C$ is the cross-correlation matrix between embeddings of different views, and $\lambda$ is a trade-off parameter.

Analysis of the gradient behavior for the Graph Barlow Twins loss $\mathcal{L}_{G-BT}$ with respect to $\alpha_{ij}$ reveals a pattern similar to that observed in our main proof:

$$\mathbb{E}_{\mathcal{G}_i, \mathcal{G}_j} \left[ \frac{\partial \mathcal{L}_{G-BT}}{\partial \alpha_{ij}} \right] = \begin{cases} < 0, & \text{if } \mu(\mathcal{G}_i) = \mu(\mathcal{G}_j) \\ > 0, & \text{if } \mu(\mathcal{G}_i) \neq \mu(\mathcal{G}_j) \end{cases} \tag{48}$$

This behavior can be understood as follows:

- When $\mu(\mathcal{G}_i) = \mu(\mathcal{G}_j)$, increasing $\alpha_{ij}$ primarily reduces the invariance term, leading to a negative gradient.

- When $\mu(\mathcal{G}_i) \neq \mu(\mathcal{G}_j)$, increasing $\alpha_{ij}$ primarily increases the redundancy reduction term, resulting in a positive gradient.

- The expectation over $\mathcal{G}_i$ and $\mathcal{G}_j$ ensures that this behavior holds on average across the dataset.

This gradient behavior demonstrates that the GBT loss induces effects similar to those observed in our main proof for the contrastive learning objective:

**(I)**. The invariance term encourages agreement between different views of the same graph, promoting $\alpha_{ij} > 0$ for graphs from the same manifold.

**(II)**. The redundancy reduction term discourages correlations between embeddings of different graphs, effectively promoting separation between graphs from different manifolds and encouraging $\alpha_{ij} \approx 0$ for such pairs.

This alignment in gradient behavior suggests that the Barlow Twins loss would lead to similar optimal coefficient configurations and, consequently, improved manifold separation as demonstrated in our main theorem. While the exact formulation differs due to the use of cross-correlations instead of mutual information, the underlying principle of increasing intra-manifold similarities while decreasing inter-manifold similarities remains consistent.

In practice, the choice between contrastive learning and Barlow Twins loss may depend on specific dataset characteristics and computational considerations. Both approaches are expected to yield improved manifold separation in the GIP framework, with potential for variations in performance depending on the nature of the graph data and the specific implementation details.

