# OpenReview forum: "Enhancing Graph Self-Supervised Learning with Graph Interplay"
_ICLR.cc/2025/Conference — ICLR 2025 Conference Withdrawn Submission_

### Official Review · Reviewer_GYfz · 2024-10-28

**Soundness:** 1
**Presentation:** 3
**Contribution:** 2
**Rating:** 3
**Confidence:** 4

**Summary:**

The authors focus on graph self-supervised learning, and propose a new augmentation technique, called Graph Interplay (GIP). GIP is to randomly link nodes belonging to different graphs in the same batch. The unimaginable results may show the potential of this newly self-supervised augmentation.

**Strengths:**

1. The proposed GIP augmentation is simple and intuitive, friendly for following research.
2. The quality of writing is easy to follow.

**Weaknesses:**

1. Potential to violate double-blind policy. In the repo for code, "environment.yml" file contains the following information "prefix: /home/zhaoxinjian/miniconda3/envs/sgnn", where "zhaoxinjian" implies one of authors.
2. The results shown in Table 1 is too unimaginable to convince. In IMDB-MULTI, GRACE+GIP and BGRL+GIP nearly double the baselines performances. For BGRL+GIP for IMDB-BINARY and GRACE+GIP for PROTENINS, the performances are 99.80$\pm$0.40 and 99.40$\pm$0.85, which approach 100 but baselines are far away from 100. Inversely, GIP does not bring so huge improvement over datasets in Table 2, although the task is still graph classification. I strongly suggest authors to double check if there is a leakage in Table 1.
3. The motivation of this paper is not clear. The authors argue other GSSLs overlook "peculiar and critical characteristics" of graph data. What are "peculiar and critical characteristics"? Why can they not capture these characteristics? Why can this paper realize this point? For example, DropEdge can capture the "varying connectivity of different nodes". Such motivation is so general that can be used for other papers.

**Questions:**

1. The theoretical result is strange. Intuitively, randomly connecting two graphs will involve more noisy edges than beneficial edges, since it's more possible to connect graphs from different distributions. In this case, the MI should be decreased, rather increased. In proof, the authors only assume a very specific scenario, where each graph only absorbs graphs within the same class and totally ignore the influence from other classes (Eq. 38). This setting is oversimplified and unreasonable.

---

> ### Author Response · Authors · 2024-11-16
>
> **W1. Potential to violate double-blind policy.**
>
> **A1**. Thank you for your careful review. We apologize for our carelessness and will withdraw our manuscript.
>
> **W2. The results shown in Table 1 is too unimaginable to convince. Inversely, GIP does not bring so huge improvement over datasets in Table 2, although the task is still graph classification.**
>
> **A2**. We have provided our source code and made every effort to review our code, to the best of our knowledge, did not identify any instances of data leakage. During pre-training, all GSSL methods were conducted without using label information, as validated in Figures 7 and 8 of the manuscript, which demonstrate that GIP indeed learns more discriminative representations. It is important to note that while fine-tuning does involve labels, our approach follows the same protocol as previous works, with the only difference being the allowance of inter-graph message-passing.
>
> The discrepancy in performance improvement between Tables 1 and 2 can be attributed to several factors:
> **a)** The datasets in Table 2 have a much lower average degree and simpler topology compared to those in Table 1. We also briefly analyzed the possible problems with the dataset in Figure 3, where we found that the higher the dropedge ratio the better the performance of the GSSL model, implying that the dataset is less dependent on the topology.
> **b)** As noted in previous work [1], some datasets (e.g., ogbg-bbbp) have imbalanced label distributions, while others (e.g., ogbg-bace) are balanced. We have achieved significant improvements on the balanced ogbg-bace dataset. The imbalanced graph learning setting is beyond the scope of our work.
> **c)** We used the provided scaffold split for Table 2 datasets, which introduces an OOD-like scenario, presenting additional challenges for representation learning [2,3]. The scaffold split method, which ensures distinct molecular skeletons between training and testing sets, may suppress GIP's advantage in improving inference accuracy through interplay when specific skeleton fine-tuning data is absent.
>
> [1] Deng, Jianyuan, et al. "A systematic study of key elements underlying molecular property prediction." Nature Communications 14.1 (2023): 6395.
>
> [2] Yang, Nianzu, et al. "Learning substructure invariance for out-of-distribution molecular representations." Advances in Neural Information Processing Systems 35 (2022): 12964-12978.
>
> [3] Chen, Dingshuo, et al. "Uncovering neural scaling laws in molecular representation learning." Advances in Neural Information Processing Systems 36 (2024).
>
> **W3. The motivation of this paper is not clear. The authors argue other GSSLs overlook "peculiar and critical characteristics" of graph data. What are "peculiar and critical characteristics"? Why can they not capture these characteristics? Why can this paper realize this point? For example, DropEdge can capture the "varying connectivity of different nodes". Such motivation is so general that can be used for other papers.**
>
> **A3**. Thank you for your insightful question regarding the motivation of our paper. We appreciate the opportunity to clarify our approach and its novelty.
>
> The "peculiar and critical characteristics" of graph data we refer to are the inherent non-Euclidean properties that distinguish graphs from other data types, including complex topological structures, non-uniform connectivity, and intricate relational linkages. Our intention is not to claim that existing GCL augmentations violate these characteristics, but to emphasize that GIP is specifically designed to respect and leverage these graph-specific properties.
>
> GIP is fundamentally rooted in the message-passing mechanism, which is the cornerstone for encoding graph data. This mechanism is tailored to handle the unique challenges of graph structures.  Specifically, GIP introduces inter-graph edges to create distinct views with different message flows and then optimizes representation consistency across these views using a GSSL loss. As the model is forced to output similar representations for both views despite their inconsistent information flows, it must learn to preserve information from graphs within the same manifold while impeding information from different manifolds.
>
> The key innovation of GIP lies in its ability to learn invariant representations capturing graph interplay within the GSSL framework. This approach goes beyond traditional graph augmentation techniques by focusing on how graph characteristics interact across multiple graphs to form more robust and informative representations.
>
> We acknowledge that our original manuscript may not have articulated these points clearly. We will revise our paper to better highlight these key contributions and include more theoretical and empirical analyses to substantiate our claims.
> Thank you for your valuable feedback, which will help us improve the clarity and robustness of our work.

---

> > ### Author Response · Authors · 2024-11-16
> >
> > **Q1. Intuitively, randomly connecting two graphs will involve more noisy edges than beneficial edges, since it's more possible to connect graphs from different distributions. In this case, the MI should be decreased, rather increased. In proof, the authors only assume a very specific scenario, where each graph only absorbs graphs within the same class and totally ignore the influence from other classes (Eq. 38). This setting is oversimplified and unreasonable.**
> >
> > **AQ1**. Self-supervised learning is essentially learning invariant representations, and strategies for data augmentation implicitly shape how such invariant representations are defined. SPAN[4] proposes elaborate edge perturbations, such that the graph spectrum between two augmented views varies widely, and then forces the model to learn spectral invariance features under the guidance of GSSL loss.
> > GIP shares a similar principle, where each graph in the two augmented views accepts message flow from different graphs, and in GSSL loss maximizes the consistency of all node representations in the two views, the model intuitively needs to be able to retain more information from the same manifold graph while filtering the information from the different manifold graphs.
> >
> > [4] Lin, L., & Chen, J. Spectral Augmentation for Self-Supervised Learning on Graphs. In The Eleventh International Conference on Learning Representations.

---

### Official Review · Reviewer_zefE · 2024-10-29

**Soundness:** 3
**Presentation:** 3
**Contribution:** 3
**Rating:** 5
**Confidence:** 4

**Summary:**

This paper presents a novel augmentation method for graph-level tasks by considering other graphs in the dataset.
The authors introduce a method to generate new edges between graphs, effectively creating an interplay between them.
This approach leads to a significant performance improvement on several benchmark datasets.

**Strengths:**

1. The paper is well-written and readable.
2. The method achieves significant performance gains on several benchmark datasets.
3. Extensive ablation study provides valuable insights into the effectiveness of different components of the proposed method.
4. The paper can empower GSSLs focusing on node-level tasks in graph-level tasks.

**Weaknesses:**

1. It may be debatable that the motivation for capturing "interplay" between graphs is not fully aligned with the method, which introduces arbitrary edges between graphs.
Even if the authors have provided extensive experiments, I think the results do not focus on aligning the motivation and implementations.
Also, there is no special consideration to differentiate inter-graph edges with intra-graph edges. This raises concerns about the validity of the "interplay" concept.
It would be better to explain how the random inter-edges improve meaningful interactions between graphs in the introduction or experiment sections.
Personally, I guess the role of interplay in the SSL process is to facilitate better learning of the invariant information (i.e., the intra-graph edges) by comparing the edges with inter-graph edges rather than capturing the relationships among graphs.
However, I am willing to authors refute my opinion and show that SSL can capture relationships with inter-graph edges.


2.  There are some typos and grammatical errors throughout the paper.

**Questions:**

1. In Theorem 1, the authors claim that equation (45) holds due to equation (46). However, it is unclear why the similarity between $f^{(1)}$ and $f^{(2)}$ guarantees the similarity between these functions and $f^\text{sopt}$. Since $f$ maps graphs from various manifolds, it might be challenging to ensure that the learning process converges to $f^\text{sopt}$. If f gets stuck in a local optimum, equation (45) might not hold, invalidating Theorem 1. Could the authors please clarify this point?

2. Is there any study that explains the proposed graph-interplay 'actually' captures the interaction between graphs?
For example, It would be better to explain the importance of inter-graph edges compared to intra-graph edges using GNN-explaining techniques such as [1].

3. It is observable that an inter-graph edge exists between nodes and does not have a notable difference compared to intra-graph edges.
Are there some reasons to design inter-graph edges in this way, instead of providing them with different semantics (e.g., learn inter-graph edge embeddings, learn important inter-graph edges with attention mechanisms)?

[1] Edge-Level Explanations for Graph Neural Networks by Extending Explainability Methods for Convolutional Neural Networks

---

> ### Author Response · Authors · 2024-11-16
>
> **W1. It may be debatable that the motivation for capturing "interplay" between graphs is not fully aligned with the method, which introduces arbitrary edges between graphs.**
>
> **A1**. We sincerely appreciate your thoughtful comments on the alignment between our motivation and method. We'd like to address your concerns and clarify our approach:
> 1. Alignment of motivation and implementation:
> The introduction of edges between graphs in GIP is not arbitrary, but a deliberate strategy designed to facilitate learning invariant representations through self-supervised learning. Our approach is motivated by the principle that graphs from the same manifold should share more mutual information than those from different manifolds (Assumption 1 in our manuscript).
> 2. Differentiating inter-graph and intra-graph edges:
> While we didn't explicitly differentiate inter-graph and intra-graph edges in our original presentation, this distinction is inherent in our method. Inter-graph edges create varied message flows between graphs, which is crucial for the learning process. The GSSL loss in our method maximizes the consistency of representations across two augmented views, forcing the model to retain messages from graphs within the same manifold while filtering out information from graphs in different manifolds.
> 3. Role of interplay in SSL:
> Your insight about the role of interplay in facilitating better learning of invariant information is astute. Indeed, the inter-graph edges serve as a contrast to intra-graph edges, helping the model learn more robust representations. However, we argue that this process also captures meaningful relationships among graphs. The model learns to distinguish relevant from irrelevant inter-graph connections, effectively capturing the interplay between related graphs.
>
> We acknowledge that these aspects could have been more clearly articulated in our original manuscript. In future revisions, we will provide more detailed explanations and potentially additional experiments to demonstrate how our method captures relationships with inter-graph edges.
> Thank you for your insightful feedback, which will help us improve the clarity and robustness of our work.
>
> **W2. Typos**
>
> **A2**. Thank you for your detailed review. We will carefully fix all the typos.
>
> **Q1. If $f$ gets stuck at a local optimum, equation (45) might not hold.**
>
> **AQ1.** Thank you for your insightful question regarding Theorem 1. Theorem 1 assumes convergence to a global optimum, which may not always be achievable in practice due to the complexity of the optimization landscape and the limitations of our algorithms.
> In practice, as you rightly point out, it is possible to fall into a local optimum, especially when the GNN encoder is not sufficiently expressive. In such cases, equation (45) might not hold, potentially limiting the practical applicability of Theorem 1.
> To address this:
> 1. We will clarify in the paper that Theorem 1 represents an ideal case and discuss its practical limitations.
> 2. We plan to conduct empirical studies to investigate how closely our method approximates the theoretical optimal solution under various conditions.
>
> We appreciate your critical analysis, as it helps improve the rigor and practical relevance of our work. We will revise our manuscript to provide a more nuanced discussion of the theoretical results and their practical implications.
>
> **Q2. Is there any study that explains the proposed graph-interplay 'actually' captures the interaction between graphs?**
>
> **AQ2**. Thank you for this insightful question. Your suggestion to use GNN-explaining techniques is valuable. While techniques like GNN-explainers exist, they are typically applied to simpler, more interpretable scenarios such as "house"-structured motif datasets or benzene ring tests. Extending these techniques to the complex scenarios our model addresses would require significant additional effort and may not provide sufficiently reliable insights due to the increased complexity. Such an investigation, while potentially valuable, is beyond the scope of our current work and remains an open direction for future research.
>
>
> **Q3.** Are there some reasons to design inter-graph edges in this way, instead of providing them with different semantics (e.g., learn inter-graph edge embeddings, learn important inter-graph edges with attention mechanisms)?
>
> **AQ3**. Thank you for this astute observation. We chose the current design for simplicity and consistency in message-passing. However, we recognize the potential benefits of differentiating inter-graph edges more explicitly. Your suggestions about learning inter-graph edge embeddings and using attention mechanisms are excellent. We will seriously consider these approaches for future iterations to enhance the expressiveness and interpretability of our model.
> We sincerely appreciate your thoughtful feedback, which will guide us in refining and extending our research.

---

### Official Review · Reviewer_KjwK · 2024-11-02

**Soundness:** 2
**Presentation:** 3
**Contribution:** 2
**Rating:** 5
**Confidence:** 5

**Summary:**

This work introduces a framework called Graph Interplay (GIP) to enhance Graph Self-Supervised Learning (GSSL) by allowing direct communication between graphs. GIP adds random inter-graph edges within a batch, enabling graphs to share information and improve representation learning. This interaction facilitates better separation of data manifolds, enhancing downstream task performance across benchmarks. Empirical results demonstrate that GIP integrates seamlessly with various GSSL methods, achieving significant performance gains.

**Strengths:**

1. The paper is well written, clearly showing the model framework.

2. Theoretical analysis based on manifold separation is sound.

3. The empirical results on graph classification demonstrate the effectiveness of the proposed model.

**Weaknesses:**

1. The model treats batched graphs as super-nodes within a larger graph structure, employing a node-level contrastive objective to learn individual representations. While this approach is straightforward and potentially powerful, the technique of pooling graphs or subgraphs into super-nodes is not novel.
2. The paper does not clearly delineate how the GIP mechanism integrates with existing GSSL models. For instance, in the context of GRACE, it is unclear whether feature or edge perturbations should still be applied to the generated views after adding inter-graph connections. Does the graph augmentation method, such as diffusion used in MVGRL, still apply to the views within the GIP framework? If not, could combining GIP's augmentation with those used in the backbone GSSL models potentially enhance performance further?
3. In Line 53, the text states that the approach is 'tailored to respect and leverage the unique attributes of graph structures,' and Line 49 mentions 'non-uniformity, varying connectivity of different nodes, and the complexity of their relational linkages.' However, the manuscript does not clearly delineate how GIP preserves these characteristics. It would be beneficial for the authors to provide both theoretical and empirical analysis to substantiate this claim.

**Questions:**

Please see the questions above.

---

> ### Author Response · Authors · 2024-11-16
>
> **W1. The model treats batched graphs as super-nodes within a larger graph structure, employing a node-level contrastive objective to learn individual representations. While this approach is straightforward and potentially powerful, the technique of pooling graphs or subgraphs into super-nodes is not novel.**
>
> **A1**. Thank you for your comment. We appreciate the opportunity to clarify our method's core contribution.
>
> The key innovation of GIP is not treating graphs as super-nodes, but rather its ability to learn invariant representations capturing graph interplay within the Graph Self-Supervised Learning (GSSL) framework. Super-node itself, to our best knowledge, does not provide so much performance enhancement. In other words, developing "GIP + GSSL" with supporting theory is the main contribution.
>
> GIP optimizes representation consistency across two views with different message flows. This is achieved by introducing inter-graph edges to create distinct views and using a GSSL loss to align representations of these views. As the model is forced to output similar representations for both views despite their inconsistent information flows, it must learn to preserve information from graphs within the same manifold while impeding information from different manifolds. This unique mechanism enables GIP to capture meaningful graph interplay while learning invariant representations that respect graph-structured data characteristics.
>
> Thank you for your feedback, which helps us improve our work's clarity and emphasis.
>
> **W2. Does not clearly delineate how the GIP mechanism integrates with existing GSSL models.**
>
> **A2**. Thank you for suggesting that we clarify the details of how GIP can be combined with existing GSSLs, in our experiments, ADDEDGE was retained as it can be seamlessly combined with GIP. Feature masking and graph diffusion were not considered as additional data augmentations, and combining these augmentation techniques may indeed yield more robust models. We will add detailed clarifications and corresponding experiments in the next version of the manuscript.
>
> **W3. In Line 53, the text states that the approach is 'tailored to respect and leverage the unique attributes of graph structures,' and Line 49 mentions 'non-uniformity, varying connectivity of different nodes, and the complexity of their relational linkages.' However, the manuscript does not clearly delineate how GIP preserves these characteristics. It would be beneficial for the authors to provide both theoretical and empirical analysis to substantiate this claim.**
>
> **A3**. We sincerely appreciate your thoughtful comment regarding lines 49 and 53. We're grateful for the opportunity to clarify our claims and provide a more detailed explanation of our approach.
>
> In our manuscript, we aim to emphasize that GIP's design is fundamentally rooted in the message-passing mechanism, which is the cornerstone for encoding graph data. This mechanism is specifically designed to handle the unique challenges of graph structures, including non-uniformity, varying connectivity of different nodes, and complex relational linkages.
>
> The distinctive nature of graph structures allows for the establishment of fine-grained message-passing between different graphs. Building on Assumption 1 in our manuscript, which posits that the mutual information between graphs from the same manifold is expected to be higher than that between graphs from different manifolds, our approach optimizes the consistency of representations of two views under different message flows. This optimization process guides the model to learn invariant representations that preserve information from graphs within the same manifold while impeding information from graphs in different manifolds.
>
> We want to emphasize that GIP is a solution tailored specifically for graph data, rather than an approach adapted from image or natural language processing techniques. This specificity is what we intended to convey in lines 49 and 53.
> We acknowledge that our original manuscript may not have articulated these points as clearly as intended. In response to your valuable feedback, we will enhance our manuscript with both theoretical and empirical analyses to better substantiate our claims and demonstrate how GIP preserves and leverages the unique characteristics of graph structures.
>
> Thank you again for your insightful comments, which will help us improve the clarity and robustness of our work.

---

### Official Review · Reviewer_5vTF · 2024-11-03

**Soundness:** 2
**Presentation:** 3
**Contribution:** 2
**Rating:** 5
**Confidence:** 3

**Summary:**

The goal of the paper is to broadens the contextual landscape within which the learning model operates. And this paper proposes a graph interplaly (GIP) framework, which encourages inter-graph connectivity fro enriched learning experiences by interconnecting graphs within learning batches. Meanwhile, the paper theoretically show the GIP could perform manifold separation via inter-graph message passing.

**Strengths:**

Pros:
1. The paper tries to broaden the contextual landscape within which the learning model operates and proposes a new graph interplay framework.
2. The framework is simple and easy to follow. The paper also provide the theory analysis to show that GIP essentially performs a principled manifold separation via combining intergraph message passing.
3. As the paper claims, the proposed framework can be readily integrated into a GSSL framework.

**Weaknesses:**

Cons:
1. Is it p a hyper-parameter of the framework? Why create enhanced views of the graph through the stochastic inter-graph edges? Could use the fully-connected graph with different edge weights to instead?
2. The paper construct inter-graph within a batch? Is it the batch size an important hyper-parameter? How the batch size impact the model performance?
3. There are also some related work about inter-graph construction, the model should highlight the difference and the improvement compared with these methods (i.e., Semi-Supervised Graph Classification: A Hierarchical Graph Perspective).
4. It is better to review the related works about inter-graph construction in the related work section.
5. The model is focused on the graph classfication. Could the framework has the improved performance on the other graph tasks, i.e., node classfication, or other settings like semi-supervised graph learning.
6. It is suggested to give the complexity analysis about the framework, i.e., time and space complexity analysis.

**Questions:**

Please see the above weaknesses.

---

> ### Author Response · Authors · 2024-11-16
>
> **W1. Is it $p$ a hyper-parameter of the framework? Why create enhanced views of the graph through the stochastic inter-graph edges? Could use the fully-connected graph with different edge weights to instead?**
>
> **A1**. Thank you for your insightful questions regarding our framework.
> Yes, $p$ is a hyperparameter of our framework.
>
> Similar to the motivation of various data augmentation methods based on probabilistic perturbation of graph structures[1,2], randomly adding edges between graphs helps to learn invariant representations. Intuitively, by optimizing the consistency between the two views augmented by GIP, we enable the model to generate similar outputs across different information streams. This approach requires the model to develop a sophisticated ability to retain messages from the same semantic streams while effectively filtering out messages from disparate or irrelevant streams.
>
> We acknowledge that a fully-connected graph with different edge weights is indeed a potentially viable alternative approach. However, such a method would incur high computational complexity in computing attention and performing dense graph message-passing. Moreover, directly applying fully connected graphs to both views would likely lead to mode collapse (where both views learn the same coefficients, making the loss optimal), necessitating additional design efforts to effectively integrate such an approach with GSSL. Our method of packing batch graphs into a large graph creates an inherently sparse graph structure, which enables more efficient computation and faster runtime performance.
>
> We appreciate your suggestion and will consider discussing these trade-offs more explicitly in our revised manuscript.
>
> [1]  Lin, L., & Chen, J. Spectral Augmentation for Self-Supervised Learning on Graphs. In The Eleventh International Conference on Learning Representations.
>
> [2] Zhu Y, Xu Y, Yu F, et al. Deep graph contrastive representation learning[J]. arXiv preprint arXiv:2006.04131, 2020.
>
> **W2. The paper construct inter-graph within a batch? Is it the batch size an important hyper-parameter? How the batch size impact the model performance?**
>
> **A2**. We appreciate your insightful question about the impact of batch size on our model's performance. You're correct in identifying batch size as an important hyperparameter, especially given our inter-graph construction within batches.
> To address your concern, we conducted additional experiments to investigate the effect of batch size on model performance. We agree that this analysis provides valuable insights, and we plan to include a more comprehensive study in the next version of our manuscript.
> As a preliminary example, we tested different batch sizes on the IMDB-MULTI dataset, using hyperparameters obtained through randomization: ($p_1$, $p_2$) = (0.8444218515250481, 0.7579544029403025). Here are our findings:
> | Batch Size | 8       | 16      | 32      | 64      | 128     | 256     |
> |------------|---------|---------|---------|---------|---------|---------|
> | Accuracy   | 59.20 ± 2.90 | 70.40 ± 1.77 | 73.73 ± 2.33 | 84.53 ± 1.76 | 91.60 ± 1.50 | 92.53 ± 0.98 |
>
> These results demonstrate a clear trend: increasing the batch size generally leads to improved performance and reduced variance. This improvement is particularly notable as we move from smaller batch sizes (8, 16, 32) to larger ones (64, 128, 256).
> We hypothesize that larger batch sizes allow for more diverse inter-graph connections, potentially leading to more robust and generalized representations.
>
> **W3 & W4. Related works about inter-graph construction.**
>
> **A3 & A4**.
> Thank you for your suggestion. We will add a discussion of inter-graph construction related works such as [3,4,5] to the next version of the manuscript.
>
> [3] Li, Jia, et al. "Semi-supervised graph classification: A hierarchical graph perspective." The World Wide Web Conference. 2019.
>
> [4] Zhao, Haihong, et al. "All in one and one for all: A simple yet effective method towards cross-domain graph pretraining."KDD 2024.
>
> [5] Zhen, Zhiwei, et al. "United We Stand, Divided We Fall: Networks to Graph (N2G) Abstraction for Robust Graph Classification under Graph Label Corruption." Learning on Graphs Conference. PMLR, 2024.

---

> > ### Author Response · Authors · 2024-11-16
> >
> > **W5. The model is focused on the graph classfication. Could the framework has the improved performance on the other graph tasks, i.e., node classfication, or other settings like semi-supervised graph learning.**
> >
> > **A5**. Thanks for your suggestion. Regarding semi-supervised graph learning, we agree that this is a valuable setting to evaluate. We appreciate your suggestion and plan to extend our experiments to include this scenario in our future work.
> > As for the node classification task, the current formulation of GIP is specifically designed for graph-level tasks, where each graph is treated as an instance. The theories and intuitions underlying GIP are not directly applicable to node-level tasks. Extending GIP to node classification would require significant modifications to the framework and additional theoretical development.
> >
> > We acknowledge that exploring these additional applications could broaden the impact of our work. In our revised manuscript, we will discuss these potential extensions as directions for future research.
> > Thank you for your valuable feedback, which helps us to consider the broader applicability of our approach.
> >
> > **W6. It is suggested to give the complexity analysis about the framework, i.e., time and space complexity analysis.**
> >
> > **A6**. Thank you for your suggestion. The amortized time complexity of GIP is $O(p * |E|)$. The space complexity is $O((1+p) * |E|+|V|)$. We will add the time complexity and space complexity analysis along with empirical runtimes in the next version of the manuscript.

---

### Official Review · Reviewer_x1Uy · 2024-11-05

**Soundness:** 3
**Presentation:** 3
**Contribution:** 3
**Rating:** 6
**Confidence:** 4

**Summary:**

This paper proposes a brand-new data augmentation, GIP, to address the limitations of current Graph Contrastive Learning augmentations which overlook the characteristics of graph. Specifically, GIP performs inter-graph edge adding instead of the common practice of edge adding or dropping within each graph, successfully improving manifold separation. The proposed method exhibits significant performance gain over existing GCL methods on TU datasets and OGB datasets. The paper provides valuable insights into future GCL augmentation designs that is more suitable for the nature of graph-structured data.

**Strengths:**

Novelty: the main contribution of the paper is to investigate inter-graph augmentation in GCL for the first time, which is simple, effective and flexible. This paper is likely to be a ground-breaking work that provides a new perspective to the GCL community.
Quality: the writing is easy to follow; the experimental results are extensive and impressive; the authors support their idea by theoretical analysis.
Reproducibility: I confirm that the authors have provided detailed code to reproduce the results.

**Weaknesses:**

1. Motivation: I do admit that inter-graph augmentation sounds interesting, but I think the motivation behind is still obscure due to the brief demonstration explaining it. The authors argued that conventional GCL augmentations overlook the characteristics of graph. While in GIP, edges between atoms of different molecules are added, which is still hard to interpret. The following questions arise: 1) How do the authors define “the peculiar and critical characteristics of graph data” in a more specific way, and how existing GCL augmentations violate it? 2) Upon the answer to Q1, how does GIP capture these characteristics?
2. Theoretical Proof: I’ve roughly checked the proof in the paper while finding it a bit confusing. For instance, to the best of my knowledge, the mathematical part before Eq.44 tries to tell that “An ideal coefficient learnt by an SSL model is better than coefficients of any other SSL models” which seems to be trivial and unnecessary; but the subsequent sentence “Since $f^opt$ represents the ideal case for GIP” is not a formal and valid proof that connects GIP to the context. Moreover, the augmentation of GIP is not involved in the proof. In conclusion, I assume that the theoretical part needs clarification to support the manifold separation assumption, which is the key theory of inter-graph augmentation in GIP.

**Questions:**

Besides the clarification of your theoretical analysis, why are some of the experimental results so high, even reaching 99% accuracy? Is there any explanation or insights? Or are there potential data leakage?

---

> ### Author Response · Authors · 2024-11-16
>
> **W1.  1) How do the authors define “the peculiar and critical characteristics of graph data” in a more specific way, and how existing GCL augmentations violate it? 2) Upon the answer to Q1, how does GIP capture these characteristics?**
>
> **A1**.  Thank you for your insightful questions. We appreciate the opportunity to clarify our approach.
> 1. We define "the peculiar and critical characteristics of graph data" as the inherent non-Euclidean properties that distinguish graphs from other data types like images or text. These include complex topological structures, non-uniform connectivity, and intricate relational linkages. Unlike grid-structured data, graphs naturally lend themselves to message-passing as a fundamental encoding method. Our intention is not to claim that existing GCL augmentations violate these characteristics, but to emphasize that GIP is specifically designed to respect and leverage these graph-specific properties as well as the message-passing mechanism.
> 2. GIP captures these characteristics by being fundamentally rooted in the message-passing mechanism, which is tailored to graph structures. Our approach introduces inter-graph connections that align with graphs' varying connectivity and relational complexity. By optimizing representation consistency across different message flows, GIP encourages the model to learn how to retain invariant information. This process helps preserve messages from graphs within the same manifold while filtering out messages from graphs in different manifolds, ultimately leading to more robust and meaningful graph representations.
> We will enhance our manuscript to articulate better how GIP leverages these unique characteristics of graph structures and demonstrate its effectiveness through both theoretical and empirical analyses.
> Thank you for your valuable feedback, which will help us improve the clarity and robustness of our work.
>
> **W2. The mathematical part before Eq.44 tries to tell that “An ideal coefficient learnt by an SSL model is better than coefficients of any other SSL models” which seems to be trivial and unnecessary. Since $f^{opt}$ represents the ideal case for GIP” is not a formal and valid proof that connects GIP to the context. Moreover, the augmentation of GIP is not involved in the proof.**
>
> **A2**. Thank you for your insightful comment. The purpose of the mathematical exposition before Eq.44 is not to make a trivial claim about the superiority of an ideal coefficient. Instead, we aim to elucidate the fundamental mechanism of GIP under the guidance of the GSSL loss. Our approach optimizes the consistency of representation output under two different information flow views. This optimization process is designed to encourage the model to:
> 1. Retain more information flow from graphs within the same manifold
> 2. Suppress information flow from graphs in different manifolds
>
> We use coefficients as an abstraction to represent the degree of information flow retention. This formulation allows us to mathematically express how GIP learns to distinguish between relevant and irrelevant information flows. Ultimately, this process leads to the learning of invariant representations that capture the essential aspects of graph interplay. By focusing on the interactions between graphs, GIP is able to generate robust and meaningful graph representations that are consistent across different views while incorporating the valuable information derived from inter-graph relationships.
>
> We acknowledge that using the ideal case does make the theory less formal, and we will refine this in the revised manuscript and attempt to incorporate data augmentation into the proofs.

---

> > ### Author Response · Authors · 2024-11-16
> >
> > **Q1. Are there potential data leakage?**
> >
> > **A1**. Thank you for your question. We have made every effort to review our code, and to the best of our knowledge, did not identify any instances of data leakage. During pre-training, all GSSL methods were conducted without using label information, as validated in Figures 7 and 8 of the manuscript, which demonstrate that GIP indeed learns more discriminative representations.
> > It is important to note that while fine-tuning does involve labels, our approach follows the same protocol as previous works, with the only difference being the allowance of inter-graph message-passing.
> >
> > Regarding the near-perfect classification performance, we highlight two potential factors:
> > 1. The direct allowance of message-passing between graphs likely simplifies graph label inference compared to GNNs that rely on parameter sharing and isolate graph label inference. This could explain the high accuracy observed in our experiments.
> > 2. The standard setting of graph classification for GSSL employs 10-fold cross-validation with a large number of training labels in the fine-tuning phase of graph classification. In contrast, the standard practice for node classification involves only 10% labeled data. We speculate that this discrepancy might reflect a trade-off strategy, suggesting that GSSL potentially encounters challenges in graph classification and may lack some critical design elements for effectively addressing graph-level learning impediments.
> >
> > We will provide more comprehensive analytical experiments and explanations in the revised manuscript to enhance the persuasiveness of our research.

---

### Author Response · Authors · 2024-11-16

We have decided to withdraw our manuscript after the discussion period, due to an oversight in anonymization, failing to fully desensitize the prefix field of environment.yml in the repository, which potentially violates the double-blind review policy. We had prepared responses to the reviewers' comments and questions regarding our manuscript. We remain committed to addressing all the valuable suggestions provided by the reviewers in a future revision of our work.

Sincere gratitude is extended to all reviewers for their time, effort, and constructive feedback, which have been valuable in improving the manuscript's quality. We also appreciate the area chair for their diligent organization of the review process.

---

> ### Author Response · Authors · 2024-11-23
>
> Dear Reviewers,
>
> Although we plan to withdraw our manuscript before the end of the discussion period, we would greatly appreciate your feedback on our rebuttal. Your insights would be invaluable for improving our work.
>
> Thank you for your time and consideration.
>
> Best regards,
>
> Authors

---

### Note · Authors · 2024-12-02

**Comment:**

We have responded to all reviewer comments in our rebuttal. However, due to an oversight in anonymization as previously explained, we need to withdraw the submission.

We sincerely thank all reviewers for their valuable comments.

**Withdrawal Confirmation:**

I have read and agree with the venue's withdrawal policy on behalf of myself and my co-authors.